# All That Glitters Is Not Gold: Key-Secured 3D Secrets within 3D Gaussian Splatting

**Yan Ren, Shilin Lu & Adams Wai-Kin Kong**
Nanyang Technological University
Singapore

## Abstract

Recent advances in 3D Gaussian Splatting (3DGS) have revolutionized scene reconstruction, opening new possibilities for 3D steganography by hiding 3D secrets within 3D covers. The key challenge in steganography is ensuring imperceptibility while maintaining high-fidelity reconstruction. However, existing methods often suffer from detectability risks and utilize only suboptimal 3DGS attributes, limiting their full potential. We propose a novel end-to-end key-secured 3D steganography framework (KeySS) that jointly optimizes a 3DGS model and a key-secured decoder for secret reconstruction. Our approach reveals that Gaussian attributes contribute unequally to secret hiding. The framework incorporates a key-controllable mechanism enabling multi-secret hiding and unauthorized access prevention, while systematically exploring optimal attribute update to balance fidelity and security. To rigorously evaluate steganographic imperceptibility beyond conventional 2D metrics, we introduce 3D-Sinkhorn distance analysis, which quantifies distributional differences between original and steganographic Gaussian parameters in the representation space. Extensive experiments show that our method achieves state-of-the-art performance in 3D reconstruction while ensuring high levels of steganographic security. The framework is highly efficient and readily extensible to multi-GPU training. The code and dataset are released at https://github.com/RY-Paper/KeySS.

## 1 Introduction

Steganography (Cheddad et al., 2010; Hu et al., 2024; Yu et al., 2023) constitutes a security methodology that conceals secret information within seemingly innocuous carriers such as images (Kadhim et al., 2019; Hamid et al., 2012; Subramanian et al., 2021), text (Majeed et al., 2021; Delina, 2008; Wu et al., 2024), audio (Dutta et al., 2020; Djebbar et al., 2011; 2012) and videos (Kunhoth et al., 2023; Mou et al., 2023; Liu et al., 2019; Sadek et al., 2015), which has demonstrated widespread applications in copyright protection (Megías et al., 2021), secure digital communications (Varghese & Sasikala, 2023), and e-commerce systems (Kumbhakar et al., 2023). The rapid advancements in 3D reconstruction technologies, such as nerual radiance fields (NeRF) (Gao et al., 2022; Bian et al., 2023; Metzer et al., 2023) and 3DGS (Fei et al., 2024; Qin et al., 2024; Yu et al., 2024), have catalyzed the development of 3D steganography, which has emerged as a promising paradigm for safeguarding 3D digital assets (Girdhar & Kumar, 2018; Zhang et al., 2023; Zhou et al., 2021; Zhu et al., 2021). Analogous to conventional steganographic techniques, 3D steganography aims to ensure that both the existence and content of secret messages remain imperceptible to unauthorized observers while maintaining reliable recovery.

Despite advancements in 3D steganography with the emergence of 3DGS, existing 3DGS-based methods still face substantial limitations in practical applications. GS-Hider (Zhang et al., 2024) modifies the standard 3DGS pipeline for the secret embedding: the coupled color features are utilized to replace the standard spherical harmonics (SH) coefficients and a scene decoder is introduced to replace standard rendering, introducing deviations from the standard GS pipeline (Figure 1(a)). While this approach achieves high fidelity for both cover and secret scenes, these modifications introduce noticeable artifacts that may raise suspicion among unauthorized users, ultimately compromising the system's imperceptibility and security. WaterGS (Guo et al., 2024) enhances imperceptibility through importance-graded SH encryption and autoencoder-assisted opacity mapping.

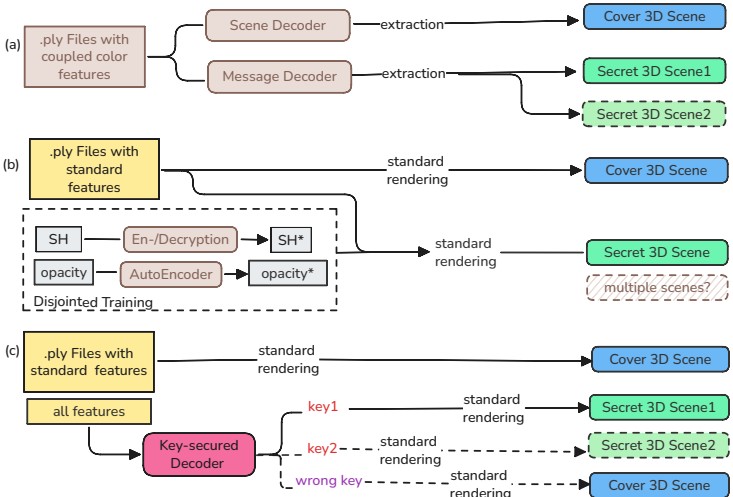

Figure 1: Compared to existing methods like (a) GS-Hider (Zhang et al., 2024) and (b) WaterGS (Guo et al., 2024), (c) KeySS maintains the standard 3DGS format compatibility while achieving superior performance through fully exploiting inherent features for fidelity and implementing a key-controllable scheme that enables both multi-secret hiding and defense against incorrect keys.

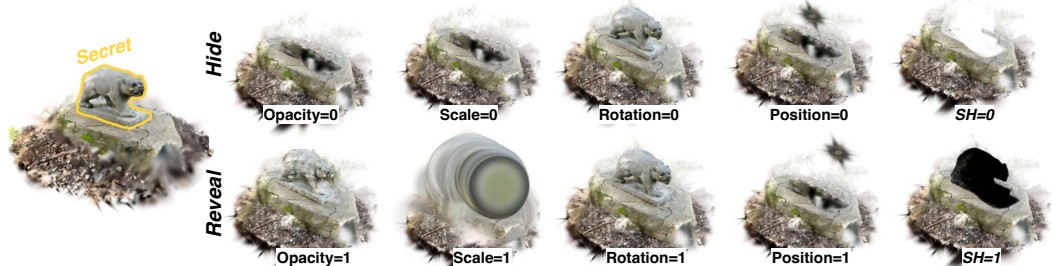

Figure 2: 3D Gaussians provide rich steganographic potential through multiple attributes: opacity, scale, rotation, position, and spherical harmonics (SH). However, naive approaches that simply zero out specific attributes to hide secrets pose fundamental security risks. The presence of hidden content can be easily detected by simply restoring the zero-value attribute, instantly revealing the hidden content. This limitation motivates our exploration of optimal for both effective hiding and security.

However, the separate concealment of SH coefficients and opacity results in a disjointed, non-end-to-end pipeline, limiting practical deployment. Additionally, the complexity of its secret embedding process prevents the encoding of multiple secret scenes within a single cover scene, reducing its flexibility and practicality in real-world scenarios.

To overcome these challenges, we introduce **Key-Secured 3D Steganography (KeySS)**, a novel framework that directly transforms cover 3D Gaussians to secret 3D Gaussians while preserving standard attribute formats and rendering processes. Our approach integrates seamlessly with existing 3DGS pipelines while providing robust security through a key-controlled mechanism without compromising visual fidelity. This key-control is critical for practical scenarios such as multi-user access, where different users retrieve distinct secrets from the same cover using unique keys, and secure content sharing, enabling selective disclosure in collaborative environments. By guaranteeing privacy and flexibility, KeySS opens opportunities for applications like personalized AR/VR experiences, secure 3D asset distribution, and controlled access in cloud-based rendering systems.

Our comprehensive analysis reveals a critical insight: **Gaussian attributes contribute unequally to steganographic effectiveness–all that glitters is not gold.** As demonstrated in Figure 2, opacity modifications effectively enable secret hiding while SH coefficients produce minimal impact or even destabilize the embedding process. However, relying solely on opacity creates a significant security

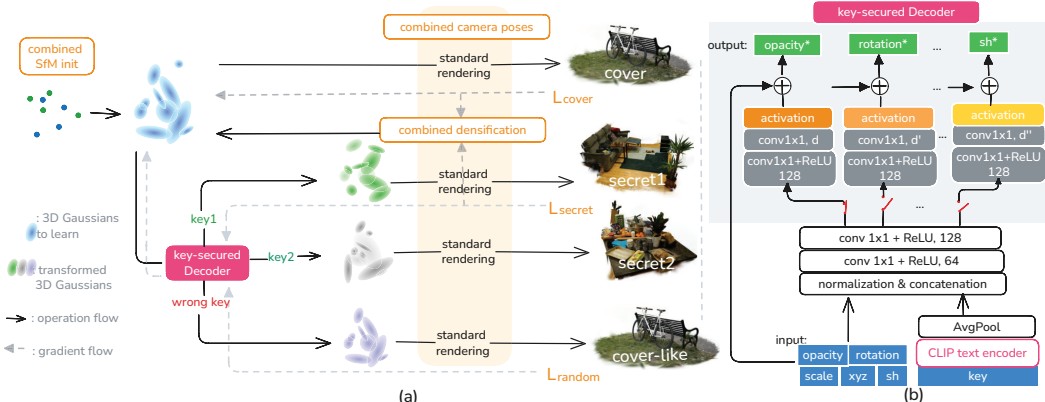

Figure 3: (a) Our end-to-end 3D steganography framework jointly trains the cover 3D Gaussians and the key-secured decoder from scratch. To enhance training, we introduce combined camera poses for diverse training samples, combined SfM points for optimal initialization, and combined densifications for refinement. (b) The key-secured decoder features a decoupled architecture with feature-specific layers for different Gaussian attributes. A key-controlled scheme enables multi-secret hiding and strengthens defenses against unauthorized extraction. Additionally, the feature-specific layers allow systematic exploration of the optimal attribute update for secret embedding.

vulnerability, as hidden information can be easily exposed by simply detecting and restoring zero-valued opacity attributes, compromising the entire steganographic system (Figure 2). Based on these findings, we systematically explore optimal attribute update combinations that strategically balance reconstruction quality and steganographic imperceptibility (Table 4). This exploration identifies that combining Gaussian attributes of opacity with rotation, position and scale significantly improves both security and fidelity over single-attribute approaches (Figure 4).

To quantitatively evaluate security beyond conventional 2D metrics, we propose a 3D-Sinkhorn security evaluation metric, which analyzes distributional differences in the Gaussian parameter space itself. Our security analysis (Table 4, Figure 7) confirms that while the opacity-only approach achieves reasonable rendering fidelity, they create distinctive statistical signatures that are invisible to conventional 2D metrics but detectable through our proposed distribution analysis, as shown in Figure 6. To summarize, KeySS makes the following contributions:

- **End-to-End 3D Steganography Framework**: An end-to-end learning framework that jointly learns cover 3D Gaussians and a key-secured decoder for 3D secret hiding, while maintaining compatibility with standard 3DGS format and rendering pipeline.

- **Key-Secured Decoder**: a key-controllable scheme enables high-fidelity multi-secret recovery while ensuring security against unauthorized access.

- **Fidelity-Security Balance Analysis**: We are the first to conduct systematic exploration of optimal 3D Gaussian attribute update combinations and introduce 3D-Sinkhorn distance as a novel security evaluation metric to balance fidelity and steganographic imperceptibility.

- **Extensive Experimental Validation**: Experimental results demonstrate that KeySS achieves superior visual quality, high reconstruction fidelity, strong robustness against unauthorized extraction, and efficient multi-GPU scalability.

## 2 PRELIMINARIES ON 3DGS

We briefly describe essential preliminaries. The rest of related work is deferred to the Supplementary. Built upon the splatting technique, 3DGS (Kerbl et al., 2023) models 3D scenes using a set of $\mathcal{N}$ anisotropic Gaussians: $\mathcal{G} = \{G_i(x)\}_{i=1}^{\mathcal{N}}$. These Gaussians are learned to capture the scene's structure and appearance, with attributes including center position $\boldsymbol{\mu} \in \mathbb{R}^3$, opacity $\alpha \in \mathbb{R}^1$, rotation $\mathbf{r} \in \mathbb{R}^4$, scale $\mathbf{s} \in \mathbb{R}^3$ and color $\mathbf{c} \in \mathbb{R}^{16 \times 3}$. Specifically, $\boldsymbol{\mu}$, $\mathbf{r}$, and $\mathbf{s}$ together describe the

configuration of the $i$-th Gaussian:

$$G_i(\mathbf{x}) = \exp\left(-\frac{1}{2}(\mathbf{x} - \boldsymbol{\mu}_i)^\top \boldsymbol{\Sigma}_i^{-1}(\mathbf{x} - \boldsymbol{\mu}_i)\right), \tag{1}$$

where $\boldsymbol{\Sigma}_i = \mathbf{R}_i \mathbf{S}_i \mathbf{S}_i^\top \mathbf{R}_i^\top$ is the 3D covariance matrix defined by the scaling matrix $\mathbf{S}_i$ and rotation matrix $\mathbf{R}_i$. Opacity $\alpha$ controls the transparency level of each Gaussian, and color $\mathbf{c}$ represents spherical harmonics (SH) to capture view-dependent appearance. For the standard rendering process from 3D to 2D, the 2D covariance matrix $\boldsymbol{\Sigma}_i'$ is formulated as $\boldsymbol{\Sigma}_i' = \mathbf{J}_i \mathbf{W}_i \boldsymbol{\Sigma}_i \mathbf{W}_i^\top \mathbf{J}_i^\top$, with the given viewing transformation matrix $\mathbf{W}$ and the Jacobian $\mathbf{J}_i$ of the affine approximation of the projective transformation. The final color of a pixel is calculated via alpha compositing:

$$C = \sum_{i \in \mathcal{N}} c_i \alpha_i' \prod_{j=1}^{i-1} \left(1 - \alpha_j'\right), \tag{2}$$

where $\alpha_i' = \alpha_i \cdot \exp\left(-\frac{1}{2}(\mathbf{x}' - \boldsymbol{\mu}_i')^\top \boldsymbol{\Sigma}_i'^{-1}(\mathbf{x}' - \boldsymbol{\mu}_i')\right)$ represents the final opacity based on the projected coordinates $\mathbf{x}'$ and $\boldsymbol{\mu}'$. Like NeRF, 3DGS initializes with SfM points and employs adaptive density control (cloning/splitting/pruning) to enhance scene detail capture.

## 3 PROPOSED KEYSS METHOD

### 3.1 PROBLEM FORMULATION

We develop an end-to-end steganographic framework (Figure 3(a)) that learns the transformation between cover and secret 3D Gaussians. More precisely, given $S + 1$ sets of ground truth 2D images $\{I_{\text{gt\_cover}}^i\}_{i=1}^{\mathcal{M}_{cover}}$ and $\{\{I_{\text{gt\_secret}}^{i,s}\}_{i=1}^{\mathcal{M}_{secret}}\}_{s=1}^{\mathcal{S}}$ with aligned camera poses, our goal is twofold: (1) learn a 3DGS model $\mathcal{G}_{\text{cover}} = \{G_i(x)\}_{i=1}^{\mathcal{N}}$ that reconstructs the cover scene, and (2) learn transformations to decode $\mathcal{G}_{\text{cover}}$ into multiple 3DGS models $\mathcal{G}_{\text{secret}}^s = \{G_j^*(x)\}_{j=1}^{\mathcal{N}}$ that render as the secret scenes. This transformation is parameterized by a decoder $D$, such that:

$$\mathcal{G}_{\text{secret}}^s = D(\mathcal{G}_{\text{cover}}). \tag{3}$$

To achieve both reconstruction fidelity and steganographic security, we jointly optimize $\mathcal{G}_{\text{cover}}$ and $D$ from scratch. This optimization is guided by a set of loss functions that enforce accurate reconstruction for both the predicted cover scene and the recovered secret scenes:

$$\mathcal{L}_{\text{cover}} = (1 - \lambda_{\text{cover}})\mathcal{L}_1(I_{\text{pred\_cover}}, I_{\text{gt\_cover}}) + \lambda_{\text{cover}}\mathcal{L}_{\text{SSIM}}(I_{\text{pred\_cover}}, I_{\text{gt\_cover}}), \tag{4}$$

$$\mathcal{L}_{\text{secret}}^s = (1 - \lambda_{\text{secret}})\mathcal{L}_1(I_{\text{pred\_secret}}^s, I_{\text{gt\_secret}}^s) + \lambda_{\text{secret}}\mathcal{L}_{\text{SSIM}}(I_{\text{pred\_secret}}^s, I_{\text{gt\_secret}}^s), \tag{5}$$

where $I_{\text{pred\_cover}}$ and $I_{\text{pred\_secret}}^s$ are rendered from $\mathcal{G}_{\text{cover}}$ and $\mathcal{G}_{\text{secret}}^s$ respectively. $\lambda_{\text{cover}}$ and $\lambda_{\text{secret}}$ are the trade-off coefficients between $L_1$ and SSIM losses.

### 3.2 KEY-SECURED DECODING ARCHITECTURE

The overview of our decoding architecture is shown in Figure 3(b). Specifically, our decoder is designed to build the transformation between the 3D Gaussians of the cover and secret scenes while maintaining a balance between fidelity and security. To further enhance security, the decoder is conditioned on a user-specific key $k^s$, ensuring that only authorized users with the correct key can accurately reconstruct the secret scenes. This can be formulated as:

$$\mathcal{G}_{\text{secret}}^s = D(\mathcal{G}_{\text{cover}}, k^s). \tag{6}$$

**The Key-Controllable Scheme** enables both multi-secret hiding capabilities and robust defense against unauthorized extraction attempts. We leverage CLIP's text encoder (Radford et al., 2021) to encode the keys, which excels at processing diverse textual inputs into semantic embeddings. The user-specific key $k$ is first tokenized and then processed through a transformer-based encoder $\mathbf{E}$ followed by average pooling operations to obtain the final key embedding: $\mathbf{k} = \text{AvgPool}(\mathbf{E}(k))$. The key embedding is concatenated with the normalized 3D Gaussian attributes as input to the decoder. To mitigate the risk of incorrect key attacks, the training process incorporates two key scenarios:

(i) correct keys for secret recovery as defined in Eqn. 6, and (ii) randomly generated incorrect keys that force the decoder to reconstruct the original cover scene:

$$\mathcal{L}_{\text{incorrect}} = (1 - \lambda_{\text{incorrect}})\mathcal{L}_1(I_{\text{pred\_incorrect}}, I_{\text{gt\_cover}}) + \lambda\mathcal{L}_{\text{SSIM}}(I_{\text{pred\_incorrect}}, I_{\text{gt\_cover}}). \quad (7)$$

The newly introduced $\mathcal{L}_{\text{incorrect}}$ ensures that when the decoder is provided with an incorrect key, it reconstructs only the cover scene without revealing any hidden information. This enforces robustness by preventing unauthorized access and strengthens the security of the steganographic system.

**The Feature-Contribution Exploration** aims to investigate the contribution of different 3D Gaussian attributes to secret hiding. The proposed decoder consists of a shared common branch and multiple feature-specific branches (Figure 3(b)). The common branch captures comprehensive representations by leveraging the full 3D Gaussian attribute space. Feature-specific decoder branches isolate distinct Gaussian attributes, enabling systematic quantification of each parameter's contribution to steganographic efficacy. Concretely, all 3D Gaussian attributes and the user-specific key are normalized and concatenated into: $\mathbf{f} = \text{concat}(\alpha, \mathbf{r}, \mathbf{s}, \boldsymbol{\mu}, \mathbf{c}, \mathbf{k})$. Then $\mathbf{f}$ is input as the common branch $\text{MLP}_{\text{common}}$ to get the common feature $\mathbf{h} = \text{MLP}_{\text{common}}(\mathbf{f})$. The common feature $\mathbf{h}$ and the Gaussian attributes of cover $\{\alpha, \mathbf{r}, \mathbf{s}, \boldsymbol{\mu}, \mathbf{c}\}$ would be passed through feature-specific branches to obtain the updated Gaussian attributes $\{\alpha^*, \mathbf{r}^*, \mathbf{s}^*, \boldsymbol{\mu}^*, \mathbf{c}^*\}$:

$$\begin{pmatrix}\alpha^* \\ \mathbf{r}^* \\ \mathbf{s}^* \\ \boldsymbol{\mu}^* \\ \mathbf{c}^*\end{pmatrix} = \begin{pmatrix}\alpha \\ \mathbf{r} \\ \mathbf{s} \\ \boldsymbol{\mu} \\ \mathbf{c}\end{pmatrix} + \boldsymbol{\theta} \circ \begin{pmatrix}\text{MLP}_{op}(\mathbf{h}) \\ \text{MLP}_{ro}(\mathbf{h}) \\ \text{MLP}_{sc}(\mathbf{h}) \\ \text{MLP}_{po}(\mathbf{h}) \\ \text{MLP}_{sh}(\mathbf{h})\end{pmatrix}, \quad (8)$$

where $\text{MLP}_{op}$, $\text{MLP}_{ro}$, $\text{MLP}_{sc}$, $\text{MLP}_{po}$, and $\text{MLP}_{sh}$ represent feature-specific branches for opacity, rotation, scale, position, and SH attributes, respectively. $\boldsymbol{\theta} \in \mathbb{R}^5$ is a binary vector that enables the selection of update combinations for the corresponding attributes. $\circ$ is the Hadamard product.

## 3.3 3D-SINKHORN EVALUATION METRIC

While fidelity can be quantified using standard image-space metrics such as PSNR, assessing steganographic security in 3D space requires a fundamentally different approach. We introduce a new security evaluation metric grounded in the Sinkhorn distance (Cuturi, 2013; Feydy et al., 2019), which measures the distributional disparities between original and steganographic 3D Gaussian parameters directly within the representation space. This approach, inspired by recent advances in optimal transport for 3D applications (Kotovenko et al., 2024), offers significant advantages over traditional 2D image-based metrics by detecting statistical anomalies in the underlying 3D representation that remain invisible in rendered views (Figure 6). The Sinkhorn distance provides an ideal balance between computational efficiency through entropic regularization and preservation of geometric correspondences critical for 3D analysis (Janati et al., 2020; Ghosal & Nutz, 2025). During evaluation, 3D Gaussians from ground truth and stego cover scenes are normalized and projected into feature-specific histograms: $g_i = \mathbf{hist}(f_i)$, $g_i^{gt} = \mathbf{hist}(f_i^{gt})$, allowing us to quantify security across different attribute distributions:

$$d = \sum_i (\mathbf{Sinkhorn}(g_i, g_i^{gt})). \quad (9)$$

By analyzing the distributional discrepancy between the ground truth cover and the stego cover in the 3D Gaussian parameter space, the 3D Sinkhorn distance metric quantifies steganographic imperceptibility at a fundamental level. Lower distributional discrepancy indicates that the stego cover maintains statistical properties nearly identical to the original, significantly enhancing resistance against both visual inspection and algorithmic detection methods. To systematically evaluate different attribute update combinations, we employ a composite score that balances reconstruction quality and statistical imperceptibility:

$$\text{score} = (\text{PSNR}_{\text{cover}} + \text{PSNR}_{\text{secret}}) \cdot (1 - d). \quad (10)$$

## 3.4 TRAINING DETAILS

**Dataset with Combined Camera Poses**: The training dataset consists of both ground truth cover images and hidden secret images, each paired with $\mathcal{M}$ corresponding camera poses. To augment

Table 1: PSNR scores for comparisons with previous works on single-secret hiding. 3DGS-GTs represent the ground truth 3DGS models used for training, serving as the theoretical upper bound for the performance of KeySS and GS-Hider respectively. The results showcase the top 3 attribute update combinations explored based on secret fidelity. For wrong key inputs, PSNR scores are evaluated against cover ("vs. cover") and secret ("vs. secret") scenes are presented to measure effectiveness of unauthorized access prevention. attributes are denoted as: opacity (op), rotation (ro), scale (sc), position (xyz), and SH (sh).

| Methods | Scene Type | Bicycle Playroom | Bonsai Counter | Room Garden | Flowers Playroom | Treehill Bicycle | Garden Playroom | Stump Playroom | Counter Bicycle | Kitchen Bonsai | Average↑ |
|---|---|---|---|---|---|---|---|---|---|---|---|
| 3DGS-GT (GS-Hider) | cover | 25.246 | 31.980 | 30.632 | 21.520 | 22.490 | 27.410 | 26.550 | 28.700 | 30.317 | 27.205 |
| 3DGS-GT (KeySS) | cover | 23.395 | 31.690 | 31.712 | 19.886 | 22.579 | 25.690 | 24.670 | 28.702 | 30.338 | 26.518 |
| 3DGS+SH | cover | 23.365 | 26.286 | 29.311 | 18.998 | 21.479 | 24.897 | 22.818 | 26.893 | 28.150 | 24.689 |
| | secret | 23.548 | 21.340 | 22.231 | 25.080 | 20.619 | 28.450 | 24.067 | 20.997 | 22.758 | 23.232 |
| 3DGS +Decoder | cover | 23.914 | 27.674 | 27.502 | 19.877 | 21.200 | 24.284 | 24.134 | 26.561 | 26.013 | 24.573 |
| | secret | 20.611 | 20.318 | 21.668 | 20.540 | 19.848 | 25.287 | 19.933 | 20.670 | 22.367 | 21.249 |
| GS-Hider (Zhang et al., 2024) | cover | 24.018 | 29.643 | 28.865 | 20.109 | 21.503 | 26.753 | 24.573 | 27.445 | 29.447 | 25.817 |
| | secret | 28.219 | 23.846 | 22.885 | 26.389 | 20.276 | 32.348 | 25.161 | 20.792 | 26.690 | 25.178 |
| KeySS (op,ro,sc,xyz) | cover | 23.011 | 31.081 | 30.785 | 19.476 | 22.433 | 25.225 | 23.827 | 28.120 | 29.862 | **25.980** |
| | secret | 29.533 | 25.456 | 23.877 | 29.272 | 22.121 | 28.179 | 29.452 | 20.891 | 29.064 | **26.427** |
| KeySS (wrong key) | v.s. cover | 22.959 | 31.023 | 30.719 | 19.625 | 22.443 | 25.120 | 23.763 | 28.098 | 29.495 | 25.916 |
| | v.s. secret | 11.010 | 10.190 | 9.751 | 9.624 | 12.458 | 9.843 | 11.228 | 10.373 | 8.593 | 10.341 |

our training dataset, we leverage the combined camera poses from both cover and secret scenes. For camera poses unique to either scene, we generate the corresponding ground truth images using pre-trained models: the cover 3D model (without embedding) for secret-scene poses, and vice versa.

**Initialization with Combined SfM Points**: The initialization from the SfM point cloud is crucial for learning 3D Gaussians. Our method uniquely employs a combined SfM point cloud for initialization, preserving the spatial information of both cover and secret scenes. This strategy enhances reconstruction quality by capturing the structural context of both datasets during initialization.

**Backpropagation with Tripled Losses**: Considering both fidelity and security, the overall loss of the proposed method can be summarized as:

$$\mathcal{L} = \beta_{\text{cover}}\mathcal{L}_{\text{cover}} + \sum_{s=1}^{S} \beta_{\text{secret}}^s \mathcal{L}_{\text{secret}}^s + \beta_{\text{incorrect}}\mathcal{L}_{\text{incorrect}}, \tag{11}$$

where $\beta_{\text{cover}}$, $\beta_{\text{secret}}^s$, and $\beta_{\text{incorrect}}$ balance the contribution of each loss term.

**Refinement with Combined Densification**: During backpropagation, a combined densification strategy is employed by leveraging the view-space positional gradients from both the cover and secret scenes. This strategy guides the process of cloning or splitting large Gaussians in $\mathcal{G}_{\text{cover}}$, allowing for finer control over the representation and improving the accuracy of the hidden information while preserving the integrity of the cover scene.

## 4 EXPERIMENTAL RESULTS

**Dataset and Implementation Details:** Following GS-Hider (Zhang et al., 2024), we use 9 scenes from Mip-NeRF360 (Barron et al., 2022) and 1 from Deep Blending (Hedman et al., 2018), forming 9 cover-secret pairs (Table 1). For multi-secret hiding, we create (cover, secret1, secret2) triplets from Mip-NeRF360 scenes (Table 2), demonstrating our method's ability to embed multiple secrets in complex 3D environments. We follow Mip-NeRF360's official evaluation protocol using its test split. Our results are reported on unseen novel views, adhering to the official Mip-NeRF360 test split and GS-Hider evaluation protocol. The training and testing sets contain distinct images, with the test set comprising novel views not observed during training. Our method builds on the 3DGS framework (Kerbl et al., 2023) and is fully compatible with recent advances in 3D Gaussian Splatting.

Table 2: PSNR scores for multiple-secret.

| Type | Scenes | PSNR↑ | Scenes | PSNR↑ | Scenes | PSNR↑ |
|---|---|---|---|---|---|---|
| Cover | Flower | 18.990 | Bicycle | 22.478 | Kitchen | 30.103 |
| Secret1 | Treehills | 21.038 | Bonsai | 29.031 | Counter | 26.753 |
| Secret2 | Garden | 21.870 | Room | 26.544 | Stump | 21.519 |
| Wrong Key | vs.Cover | 18.778 | vs.Cover | 16.456 | vs.Cover | 29.638 |
| | vs.Secret1 | 10.598 | vs.Secret1 | 12.117 | vs.Secret1 | 8.910 |

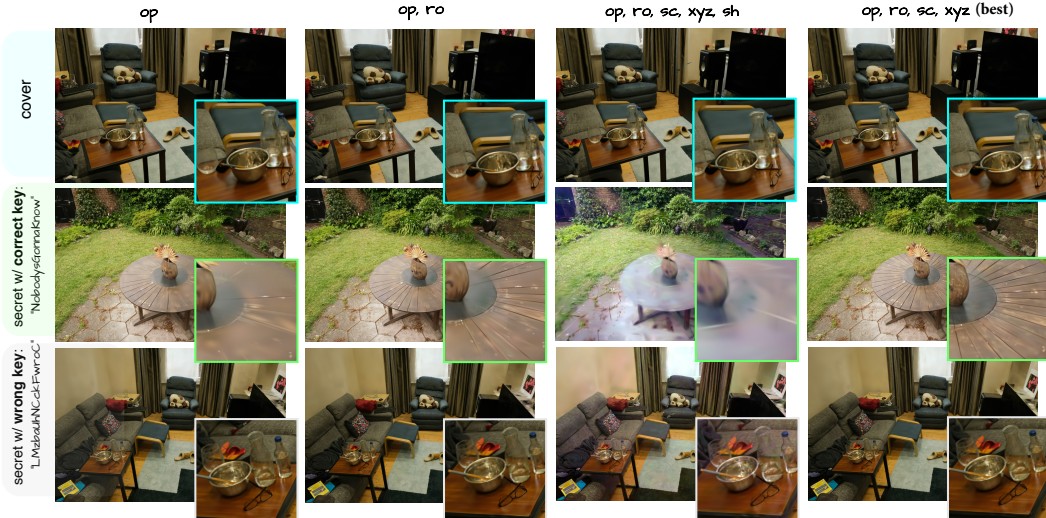

Figure 4: Visualization of decoder outputs across different attribute update combinations using correct and incorrect keys. The last two rows show secret recovery (correct key) and security preservation (incorrect key). Notation follows Table 1.

We adopt the same training hyperparameters and rendering process as 3DGS, training for 30,000 iterations on a 24GB NVIDIA RTX 6000 GPU. Due to memory limits, we cap Gaussians in the cover scene at 500,000. Ground-truth scenes are also trained with 3DGS. Loss weights $\lambda$cover, $\lambda$secret, and $\lambda_{\text{incorrect}}$ are set to 0.5, following 3DGS defaults. Balance coefficients $\beta_{\text{cover}}$ and $\beta_{\text{secret}}$ are set to 0.5, and $\beta_{\text{incorrect}}$ to 0.01, balancing reconstruction, secret recovery, and robustness against incorrect-key attacks. Keys are 16-character alphanumeric strings (uppercase and lowercase), balancing security and computational feasibility, embedded via the pre-trained CLIP ViT-L/14 model. Performance is evaluated with PSNR; SSIM and LPIPS results appear in the Supplementary.

## 4.1 FIDELITY ASSESSMENT

**Single-Secret Hiding**: We compare KeySS against the baseline GS-Hider (Zhang et al., 2024), as shown in Table 1. As the open-source implementations of the existing methods are unavailable, we directly compare our results with the reported figures from their papers. Table 1 presents the quality of our ground truth 3DGS cover, which is trained from scratch (Kerbl et al., 2023) and serves as the theoretical upper bound of our performance. However, when compared to the ground truth 3DGS used in GS-Hider, our upper bound is lower, indicating differences in the underlying training setups. Despite having a lower upper limit, KeySS achieves higher fidelity in cover reconstruction, with a minimal fidelity reduction of only 0.511 dB from the upper limit, compared to 1.388 dB in the baseline method. KeySS also excels in secret preservation, outperforming the baseline by 4.9%. More single-secret hiding visualizations are provided in Figure 4.

**Multiple-Secret Hiding**: To seamlessly embed an additional secret into the cover scene, we extend our model by simply incorporating another secret loss term, without introducing any further architectural modifications. This straigtforward extension highligts the flexibility of our approach, as it naturally scales to multiple hidden secrets without requiring structural changes or additional constraints. Table 2 presents the PSNR scores for multi-secret hiding, demonstrating that our method preserves high cover scene fidelity while maintaining competitive secret reconstruction quality, comparable to the single-secret results in Table 1. As illustrated in Figure 5, our model effectively reveals different hidden secrets based on the provided key input.

We experimentally evaluate GS-Hider, and assess WaterGS and SecureGS at the framework level, assessing imperceptibility, flexibility, and practicality, for

Table 3: Key element comparison across existing methods.

| Method | EndToEnd | 3DGS-GT | Decoder | Security | Code | MultiSecret |
|--------|----------|---------|---------|----------|------|-------------|
| WaterGS | ✗ | ✗ | AutoEncoder | StegExpose | ✗ | ✗ |
| SecureGS | ✔ | ✗ | MLPs | ✗ | ✗ | ✗ |
| GS-Hider | ✔ | ✔ | CNN | StegExpose | ✗ | ✔ |
| Ours | ✔ | ✔ | MLPs (125.7k) | 3D Sinkhorn | ✔(1st to share) | ✔ |

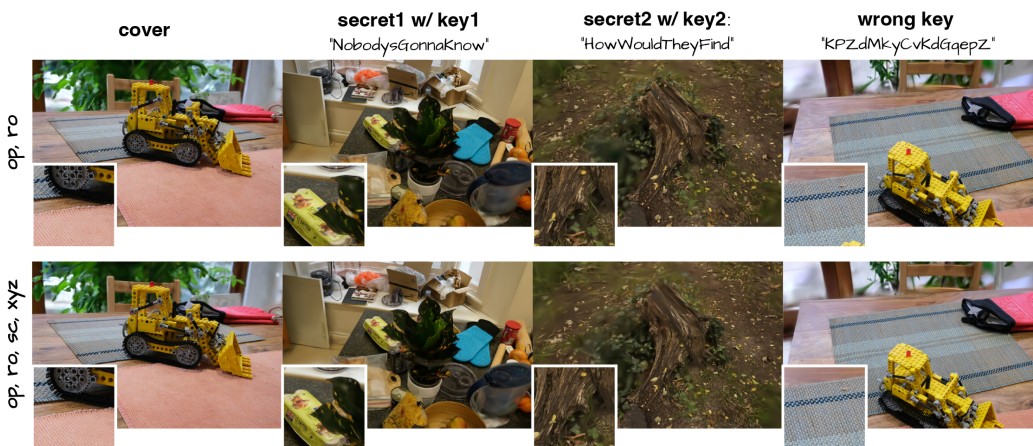

Figure 5: Visualization comparison of our method on multiple secret hiding across different attribute update scenarios using both correct and incorrect key inputs. The notation is consistent with Table 1.

the following reasons: (1) Neither method is open-sourced, making it impossible to evaluate reproducible metrics such as 3D Sinkhorn distance. (2) Reported PSNR values are not directly comparable, as the ground truth 3DGS used in their experiments are unspecified and potentially differ (Table 3).

## 4.2 SECURITY EVALUATION

**2D Steganalysis with StegExpose:** Data security is a fundamental concern in steganography, where the objective is to conceal information without drawing attention. Following prior work, we adopt the 2D steganalysis tool StegExpose (Boehm, 2014) to evaluate the detectability of stego cover scenes. A detection dataset is constructed by mixing equal proportions of cover scenes with and without hidden secrets.

The ROC curves in Figure 6 are obtained by varying StegExpose detection thresholds across a broad range. In the ideal case, the steganalyzer should perform no better than random guessing, achieving a classification probability of 50% and an AUC of 0.5, which corresponds to a diagonal ROC curve (Jing et al., 2021; Mou et al., 2023). As shown in Figure 6, our method exhibits stronger resistance to steganalysis attacks, generating stego covers that are significantly less detectable than those from existing approaches. Notably, this high level of security is preserved even when embedding multiple secrets, highlighting the framework's effectiveness in achieving both secure and imperceptible information hiding. Moreover, traditional 2D image-based metrics are insufficient for evaluating the security of 3D steganography.

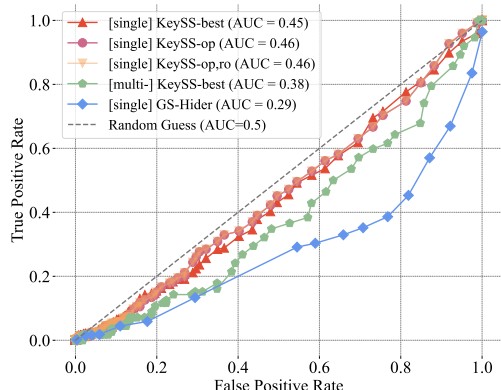

Figure 6: ROC curves from StegExpose analysis. 'single' and 'multi-' refer to single- and multi-secret hiding. 'best' denotes the optimal attribute update combination (op, ro, sc, xyz).

As shown in Figure 6, different attribute update strategies in our method produce similar StegExpose results (comparable AUC values), which fail to fully capture the robustness of the 3DGS models. This underscores the need for dedicated 3D-specific evaluation metrics, such as our proposed 3D-Sinkhorn metric.

**Security Against Unauthorized Access:** Our decoder employs a key-controllable scheme that effectively resists wrong key attacks. Table 1 (last row) reports the impact of the $\mathcal{L}_{\text{incorrect}}$ loss in preventing unauthorized secret retrieval using randomly generated incorrect keys unseen during

Table 6: Ablation study on CLIP and user key.

| Pretrained CLIP-based Encoder Variant | | | Key Length | | | PSNR (Bonsai, Counter) | |
|---|---|---|---|---|---|---|---|
| base-patch16 | large-patch14 | laion2B-s32B-b82K | 16 | 32 | 64 | Cover | Secret |
| ✔ | | | ✔ | | | 30.826 | 22.722 |
| | ✔ | | ✔ | | | 31.081 | 25.456 |
| | | ✔ | ✔ | | | 31.035 | 25.183 |
| | ✔ | | | ✔ | | 31.133 | 25.641 |
| | ✔ | | | | ✔ | 31.063 | 25.550 |

Table 7: Training details of our method.

| Type | Setting | Time | GPU Memory | Gaussians | Iterations | Cover (Bosnai) | Secret (Counter) |
|---|---|---|---|---|---|---|---|
| Single | 1GPU | 3h54m | 12GB | 500k | 30k | 31.081 | 25.456 |
| | 1GPU,4K | 4h14m | 12GB | 500k | 50k | 30.956 | 27.074 |
| | 4GPU | 18m | 19GB | 690k | 30k | 31.054 | 23.879 |
| | 4GPU,4K | 30m | 19GB | 1285k | 50k | 31.963 | 23.910 |
| Multi | 1GPU | 6h33m | 18GB | 500k | 30k | (Kitchen, Counter, Stump) | |

training. The average PSNR scores against both ground truth cover and secret scenes confirm the decoder's robustness in protecting hidden information. Table 2 extends this evaluation to multiple-secret hiding scenarios, further demonstrating security across settings. Visual examples in Figures 4 and 5 show how the decoder responds to incorrect keys, ensuring secret content remains inaccessible without the correct key. This key-controllable scheme offers flexibility, allowing seamless integration of additional losses to strengthen security against unauthorized access.

## 4.3 ABLATION STUDY

**Balancing Fidelity and Security**: The ablation study in Table 4 evaluates the performance of different attribute update combinations learned by our decoder. The top three combinations for cover and secret fidelity are: a single attribute update with opacity; a double update with opacity and rotation; and a quadruple update with opacity, rotation, scale, and position. Detailed PSNR results for these are provided in Table 1. Consistent with our intuition (see Figure 2), opacity contributes most significantly to secret hiding, while the complex, high-dimensional color/SH attributes are harder to learn and less effective for concealment. The composite score indicates that updating all four attributes offers the best balance between fidelity and security.

Table 4: Ablation study on different attribute update combinations. The top three methods are highlighted in gray.

| Count | Updated attributes | | | | | Scene PSNR | | 3D Sinkhorn↓ | Score↑ |
|---|---|---|---|---|---|---|---|---|---|
| | op | ro | sc | xyz | sh | Cover↑ | Secret↑ | | |
| 1 | ✔ | | | | | 26.020 | 26.138 | 0.181 | 42.717 |
| | | ✔ | | | | 21.752 | 23.784 | 0.268 | 33.314 |
| | | | ✔ | | | 25.080 | 20.497 | 0.190 | 36.921 |
| | | | | ✔ | | 21.980 | 23.866 | 0.211 | 36.182 |
| | | | | | ✔ | 22.825 | 12.530 | 0.162 | 29.621 |
| 2 | ✔ | ✔ | | | | 26.036 | 26.113 | 0.175 | 43.008 |
| | ✔ | | ✔ | | | 25.998 | 26.038 | 0.208 | 41.228 |
| | ✔ | | | ✔ | | 25.777 | 21.615 | 0.329 | 31.795 |
| | ✔ | | | | ✔ | 25.831 | 24.118 | 0.213 | 39.319 |
| 3 | ✔ | ✔ | ✔ | | | 25.632 | 24.743 | 0.206 | 39.987 |
| | ✔ | ✔ | | ✔ | | 25.411 | 25.662 | 0.219 | 39.899 |
| | ✔ | ✔ | | | ✔ | 24.643 | 21.645 | 0.185 | 37.729 |
| 4 | ✔ | ✔ | ✔ | ✔ | | 25.980 | 26.427 | 0.153 | **44.389** |
| | ✔ | ✔ | | ✔ | ✔ | 25.951 | 19.970 | 0.430 | 26.157 |
| 5 | ✔ | ✔ | ✔ | ✔ | ✔ | 25.832 | 20.961 | 0.256 | 34.810 |

To further assess security, Figure 7 visualizes low-opacity regions in cover scenes by marking Gaussians with opacity below 0.05 to detect potential information leakage patterns. The quadruple-attribute update achieves better steganographic imperceptibility, evidenced by a lower 3D-Sinkhorn distance. In contrast, relying solely on opacity updates results in more concentrated low-opacity areas, compromising security due to potentially detectable patterns.

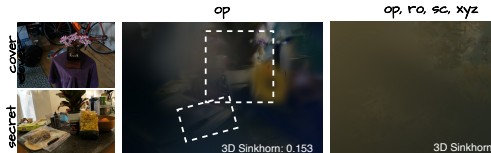

Figure 7: Low-opacity areas in the cover image.

**Combined Gaussian Optimization:** Our method integrates three combination strategies: combined SfM point clouds for initialization, combined camera poses to enrich training samples, and combined densification for refinement. Table 5 summarizes the ablation results. SfM initialization is critical, as using SfM points from only the cover or secret scene biases reconstruction toward that scene. The combined camera poses strategy balances performance by providing diverse training data for both scenes. Lastly, combined densification improves results by leveraging view-space positional gradients from both scenes during refinement.

Table 5: Ablation study on combined SfM initialization, camera poses, and densification.

| Combined SfM | Combined Camera Poses | Combined Densification | Cover | Secret | Average |
|---|---|---|---|---|---|
| cover | cover | ✔ | 22.830 | 29.719 | 26.275 |
| cover | ✔ | ✔ | 23.279 | 27.083 | 25.181 |
| ✔ | cover | ✔ | 23.300 | 26.855 | 25.077 |
| secret | ✔ | ✔ | 19.907 | 29.761 | 24.834 |
| secret | secret | ✔ | 19.789 | 29.719 | 24.754 |
| ✔ | secret | ✔ | 19.763 | 29.693 | 24.728 |
| ✔ | ✔ | cover | 22.888 | 29.611 | 26.249 |
| ✔ | ✔ | ✔ | 22.911 | 29.858 | **26.384** |

**CLIP & Key Ablation and Runtime:** We conduct an ablation study on key length and CLIP encoder variants, as summarized in Table 6. As shown in Table 7, our framework achieves efficient training and inference, and is readily scalable to multi-GPU setups. When integrated with a distributed Gaussian Splatting backbone such as Grendel-GS (Zhao et al., 2024), the system supports

4K-resolution training with 1.28 million Gaussians in just 30 minutes using 4 GPUs. At inference time, the model renders at 130 FPS per key input.

## 5 Conclusion and Limitations

In this paper, we present KeySS, an end-to-end 3D steganography framework that jointly optimizes cover 3D Gaussians and a key-secured decoder. Our decoder maintains imperceptibility by adhering to the standard 3D-GS format and rendering pipeline, while incorporating a key-controllable scheme that enables robust multi-secret hiding and strong resilience against incorrect key attacks. Task-specific decoder branches facilitate systematic exploration of optimal attribute updates for high-fidelity secret concealment. Additionally, we introduce 3D-Sinkhorn, a novel metric designed to quantify steganographic imperceptibility in 3D, overcoming the limitations of traditional 2D steganalysis metrics and paving the way for future research in 3D steganography. Extensive experiments demonstrate that KeySS achieves state-of-the-art performance in both fidelity and security.

**Limitations**:

**Trade-off Between Cover Fidelity and Secret Fidelity.** Despite strong performance, KeySS exhibits an inherent trade-off between cover fidelity and secret fidelity due to the joint optimization of 3DGS and the key-secured decoder (Table 4). Because both tasks share the same Gaussian attributes, the model cannot fully optimize cover rendering and secret embedding simultaneously. Improved objective weighting or disentangled representations may help alleviate this limitation.

**Potential Secret Leakage.** The current key-encoder design encodes a 16-character letter-only key ($52^{16}$ possibilities), while training involves only 30k iterations, preventing effective separation across the full key space. Consequently, nearby keys in the embedding space may decode correlated information, posing a risk of approximate-key leakage. This stems from treating keys as continuous embeddings rather than enforcing discrete cryptographic behavior. Future work may incorporate cryptographic mechanisms, such as hash-based key expansion or discrete key-derivation, to ensure decorrelated outputs for even small key perturbations.

**Sensitivity to Gaussian Pruning.** Secret reconstruction depends on contributions from all Gaussians, making the method sensitive to pruning. Removing Gaussians that encode secret-relevant information can degrade or eliminate the hidden message, revealing a tension between robustness and representation compression. Future extensions may focus on concentrating secret information within a small, stable subset of Gaussians, incorporating redundancy or error-correction, or training with pruning-aware constraints.

## 6 Acknowlegement

This research is supported by the National Research Foundation, Prime Minister's Office, Singapore, and the Ministry of Digital Development and Information, under its Online Trust and Safety (OTS) Research Programme (MDDI-OTS-001). Any opinions, findings and conclusions or recommendations expressed in this material are those of the author(s) and do not reflect the views of National Research Foundation, Prime Minister's Office, Singapore, the Ministry of Digital Development and Information, or the Centre for Advanced Technologies in Online Safety.

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

## 7 APPENDIX

Additional implementation details and comparisons are provided in the supplementary material. Unless stated otherwise, all experiments are conducted using our method with the optimal attribute update combination: (op, ro, sc, xyz).

## A RELATED WORK ON 3D SCENE RECONSTRUCTION

3D scene reconstruction aims to generate a 3D scene from a set of images and other data, while rendering projects 3D models into 2D images based on given camera poses. Traditional methods include structure-from-motion (SfM) (Snavely et al., 2006) and multi-view stereo (MVS) (Snavely et al., 2006) algorithms. With the rise of deep learning, NeRF (Mildenhall et al., 2021) encodes scene information by overfitting a multi-layer perceptron (MLP), enabling photorealistic novel view synthesis from limited input images. Despite revolutionizing image synthesis, NeRF suffers from high computational costs and limited controllability due to its implicit representation. 3DGS (Kerbl et al., 2023) emerges as a solution to these challenges, providing an explicit representation and highly parallelized workflows for efficient rendering and reconstruction. It represents scenes with learnable 3D Gaussians, which are projected onto image planes through a splatting process, enabling high-quality rendering with real-time performance. Our method builds upon the strengths of 3DGS in both reconstruction and rendering while leveraging the high-capacity embedding potential of millions of 3D Gaussians to conceal the secret scenes effectively.

## B RELATED WORK ON 3D STEGANOGRAPHY

3D steganography, using 3D models to hide secret messages, has been studied for decades (Zhang et al., 2023; Zhou et al., 2021; Zhu et al., 2021; Girdhar & Kumar, 2018). Traditional methods primarily modify 3D mesh geometry or topology for copyright protection, such as adjacent bin mapping (Wu & Dugelay, 2009), triangle mesh reformation (Thiyagarajan et al., 2013), and vertex decimation (Tsai, 2014). Other approaches retrieve hidden messages from 2D renderings of 3D distortions (Yoo et al., 2022). Recent advances in 3D reconstruction have shifted focus to more powerful representations, like NeRF and 3DGS, enabling new possibilities for steganography in neural rendering. For example, CopyNeRF (Luo et al., 2023) embeds copyright protection into NeRF models using watermarked color representations and a resistant rendering scheme, while WateRF (Jang et al., 2024) applies discrete wavelet transformation to both implicit and explicit NeRF models. NeRFProtector (Song et al., 2024) introduces a plug-and-play strategy for protecting NeRF copyrights. However, these methods are primarily focused on embedding binary bit messages (Lu et al., 2025) with limited hiding capacity. StegaNeRF (Li et al., 2023) pioneers higher-capacity hiding, including images and audio, within NeRF rendering. More recently, GS-Hider (Zhang et al., 2024) and WaterGS (Guo et al., 2024) have explored hiding 3D content in 3D reconstruction models. GS-Hider replaces 3DGS's SH coefficients with coupled secure features and introduces separate decoders for the cover and secret scenes. However, these non-standard features and rendering process risk arousing suspicion and violating the imperceptibility principle of steganography. WaterGS, aligned with standard 3DGS rendering, uses importance-graded SH coefficient encryption and opacity mapping for secret embedding. However, by focusing primarily on SH and opacity attributes, it underutilizes the full 3DGS attributes and lacks end-to-end training, limiting flexibility in embedding multiple secrets in a single cover. Our proposed method, KeySS, aims to develop an end-to-end learnable 3D steganography framework that has both imperceptibility and flexibility.

## C IMPLEMENTATION DETAILS

The proposed decoder (Figure 3 (b)) consists of a shared common branch and multiple feature-specific branches, ensuring both simplicity and efficiency. The common branch comprises two MLP layers with ReLU activation for general feature extraction. The common branch receives concatenated features derived from L2-normalized Gaussian attributes as input. Each feature-specific branch contains two additional MLP layers followed by feature-specific activation functions. To enhance training stability and facilitate gradient flow, residual connections are incorporated. For the MLP architecture, we employ 1×1 convolutional layers instead of standard linear layers, which is motivated by their effectiveness in PointNet (Qi et al., 2017). The use of 1×1 convolutions enables efficient

Table 8: Comparison of Training and Testing Results with PSNR and View Counts

| Methods | Scene Type | Bicycle Playroom | Bonsai Counter | Room Garden | Flowers Playroom | Treehill Bicycle | Garden Playroom | Stump Playroom | Counter Bicycle | Kitchen Bonsai | Average |
|---|---|---|---|---|---|---|---|---|---|---|---|
| training views (count) | cover | 255 | 255 | 272 | 151 | 123 | 161 | 109 | 210 | 169 | 189.444 |
| | secret | 196 | 210 | 161 | 196 | 169 | 196 | 196 | 169 | 255 | 194.222 |
| novel testing views (count) | cover | 25 | 37 | 39 | 22 | 18 | 24 | 16 | 30 | 25 | 26.222 |
| | secret | 29 | 30 | 24 | 29 | 25 | 29 | 29 | 25 | 37 | 28.556 |

local feature aggregation while maintaining spatial awareness, leading to improved performance in our attribute update strategy. Based on this architecture, the proposed decoder is equipped with a key-controllable mechanism for enhanced security and a selective update scheme to fully explore the hiding potential of various attribute update combinations. During training, we enforce robustness by pairing each correct key with a wrong key in every iteration, resulting in a 1:1 ratio, which we have explicitly highlight in the revised paper. This design ensures both security and practical usability.A more detailed breakdown of training and testing view counts is provided in Table 8.

## D    OVERFITTING STUDY

A potential concern is whether a shared decoder, when trained to reconstruct multiple secrets (e.g., secrets A and B) from a single cover image, might overfit to one particular secret and consequently fail to recover the others. Our results indicate that this issue does not arise in practice. As reported in Table 2 and illustrated in Figure 5, the shared decoder achieves consistently high PSNR across all embedded secrets, suggesting that it neither collapses to a single supervision signal nor preferentially reconstructs any particular secret.

This robustness primarily follows from our key-conditioned architecture. Each secret is associated with a distinct key, and the decoder is conditioned on the corresponding key embedding during reconstruction. The key thus provides a strong, discriminative signal that guides the decoder toward the appropriate secret. This mechanism is analogous to conditioning in modern class- or text-guided generative models, where a single generator successfully produces diverse outputs (e.g., images of different classes or text prompts) despite full parameter sharing. In those settings, as in ours, the conditioning input (class label, text description, or secret key) effectively partitions the task space and prevents overfitting to any individual target.

Within our framework, the key embedding steers the shared decoder to the correct reconstruction trajectory and prevents representational collapse toward a specific secret. Consequently, the decoder maintains stable performance even as the number of embedded secrets increases.

## E    GEOMETRIC CONSISTENCY

To address geometric consistency, we provide depth map visualizations of our reconstructions in Figure 8. Additionally, we will release all trained checkpoints on GitHub to ensure reproducibility and allow independent verification.

## F    RENDERING SPEED COMPARISON

As shown in Table 9, we assess the adaptability of our method and baselines within the SIBR Viewer rendering engine provided by 3DGS (Kerbl et al., 2023). Unlike GSHider, but similar to WaterGS, our approach maintains full compatibility with the standard 3DGS format and rendering process, allowing seamless integration into the original 3DGS pipeline without requiring modifications. This ensures that our method can be readily deployed in existing 3DGS-based applications without additional engineering overhead.

Furthermore, Table 9 demonstrates that our method preserves the standard rendering efficiency of 3DGS, achieving an average rendering speed of 130 FPS. This result indicates that our steganographic enhancements do not introduce significant computational overhead, maintaining real-time performance comparable to the original 3DGS framework.

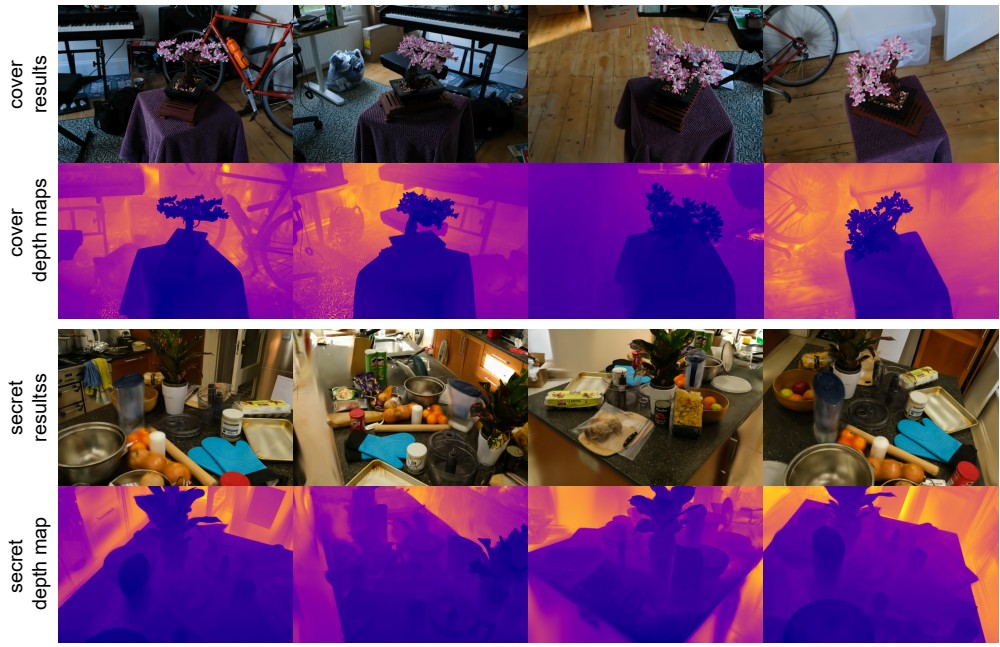

Figure 8: The depth map for cover and secret scenes in varied views.

## G    DETAILED 3D-SINKHORN RESULTS

A comprehensive breakdown of the 3D Sinkhorn distance analysis, including per-attribute comparisons, is presented in Table 10. This metric provides a fine-grained evaluation of the distributional discrepancy between the ground truth cover and the stego cover in the 3D Gaussian space, offering crucial insights into the imperceptibility of the hidden secret. As we can find, the histogram distance between the scales' histograms are very close which indicates that the scale value tends to distribe relatively equally in 3DGS models. The other attribute distribution differs more.

A comprehensive breakdown of the 3D-Sinkhorn distance analysis, including detailed per-attribute comparisons across opaciy, rotation, scale, position, and spherical harmonics, is presented in Table 10. This sophisticated metric quantifies the distributional discrepancy between the ground truth cover and the stego cover in the 3D Gaussian parameter space, offering crucial insights into the imperceptibility of the hidden secret. The analysis reveals a notable pattern: the histogram distances between scale distributions are consistently minimal across all tested scenes, indicating that scale parameters tend to distribute relatively uniformly in well-optimized 3DGS models regardless of content. In contrast, other attribute distributions, particularly opacity and rotation, exhibit higher variability between cover and stego models. This disparity suggests that these attributes provide more exploitable degrees of freedom for secret embedding while maintaining perceptual fidelity, aligning with our quantitative performance results. The 3D-Sinkhorn analysis thus provides statistical validation for our attribute update combination strategy, confirming that the method optimally utilizes the available feature space for steganographic purposes.

## H    ADDITIONAL QUANTITATIVE RESULTS

Due to the unavailability of public codebases for existing 3D steganography methods (Zhang et al., 2024; Guo et al., 2024), we independently train the ground-truth cover and secret scenes from scratch using the original 3DGS framework (Kerbl et al., 2023). This approach ensures a fair baseline for comparison while maintaining consistency with standard 3DGS optimization procedures and rendering pipelines. The quality metrics (PSNR, SSIM, and LPIPS) of the original scenes are comprehensively documented in Table 11. Compared with the original scene performance reported by GS-Hider, our baseline models show lower scores across all 3 metrics. This performance gap in

Table 9: Comparison of the average rendering speed and the adaptability of SIBR viewer (Kerbl et al., 2023).

| Methods | Scene Type | SIBR Viewer | Averaged Rendering FPS |
|---|---|---|---|
| 3DGS (Kerbl et al., 2023) | cover | ✔ | 130 |
| GS-Hider (Zhang et al., 2024) | cover | ✘ | 45 |
| | secret | ✘ | 45 |
| WaterGS (Guo et al., 2024) | cover | ✔ | 100+ |
| | secret | ✔ | 100+ |
| KeySS | cover | ✔ | 130 |
| | secret | ✔ | 130 |

Table 10: Detailed breakdown of 3D-Sinkhorn distances.

| Count | Updated attributes | | | | | 3D Sinkhorn Distance↓ | | | | | |
|---|---|---|---|---|---|---|---|---|---|---|---|
| | op | ro | sc | xyz | SH | $\sum$ | op | sc | ro | xyz | SH |
| 1 | ✔ | | | | | 0.181 | 0.079 | 4e-4 | 0.030 | 0.038 | 0.034 |
| | | ✔ | | | | 0.268 | 0.133 | 5e-4 | 0.050 | 0.042 | 0.043 |
| | | | ✔ | | | 0.190 | 0.060 | 5e-4 | 0.041 | 0.042 | 0.047 |
| | | | | ✔ | | 0.211 | 0.099 | 4e-4 | 0.031 | 0.046 | 0.034 |
| | | | | | ✔ | 0.162 | 0.066 | 4e-4 | 0.030 | 0.040 | 0.026 |
| 2 | ✔ | ✔ | | | | 0.175 | 0.063 | 5e-4 | 0.038 | 0.034 | 0.040 |
| | ✔ | | ✔ | | | 0.208 | 0.093 | 5e-4 | 0.038 | 0.044 | 0.033 |
| | ✔ | | | ✔ | | 0.329 | 0.084 | 4e-4 | 0.101 | 0.073 | 0.071 |
| | ✔ | | | | ✔ | 0.213 | 0.095 | 4e-4 | 0.042 | 0.041 | 0.034 |
| 3 | ✔ | ✔ | ✔ | | | 0.206 | 0.084 | 5e-4 | 0.033 | 0.054 | 0.036 |
| | ✔ | ✔ | | ✔ | | 0.219 | 0.102 | 5e-4 | 0.048 | 0.043 | 0.026 |
| | ✔ | ✔ | | | ✔ | 0.185 | 0.077 | 4e-4 | 0.030 | 0.041 | 0.037 |
| 4 | ✔ | ✔ | ✔ | ✔ | | 0.153 | 0.052 | 5e-4 | 0.027 | 0.037 | 0.036 |
| | ✔ | ✔ | | ✔ | ✔ | 0.430 | 0.301 | 5e-4 | 0.038 | 0.038 | 0.053 |
| 5 | ✔ | ✔ | ✔ | ✔ | ✔ | 0.256 | 0.125 | 5e-4 | 0.035 | 0.037 | 0.058 |

the ground truth models should be considered when interpreting the steganographic results, as it indicates our method starts from a lower baseline across these metrics.

Comprehensive quality metrics (PSNR, SSIM, and LPIPS) for our method with the optimal attribute update combination are presented in Table 12. The results demonstrate that our approach achieves strong rendering performance, with only a minimal PSNR reduction of 0.78 dB compared to the original 3DGS baseline in Table 11. While our SSIM and LPIPS scores appear lower than those reported by GS-Hider, this discrepancy primarily stems from our ground truth models starting from a lower baseline across these metrics. When evaluating the relative performance degradation from their respective baselines, our method demonstrates comparable reduction margins to GS-Hider. This suggests that despite operating under more challenging baseline conditions and incorporating additional security features, our steganographic approach preserves visual quality at a similar level. As previously mentioned, our method can be extended to other advanced 3DGS models, ensuring broader applicability and compatibility with recent developments in the field.

## I  ROBUSTNESS OF THE DECODER

To rigorously evaluate the robustness of our proposed decoder architecture, we conducted ablation studies comparing performance with and without user-specific key integration. As shown in Table 13, the decoder maintains comparable performance levels when incorporating user-specific keys relative to the baseline model without keys. This demonstrates that our key integration mechanism successfully enables multi-scene hiding capabilities and wrong key defense functionality without compromising the decoder's reconstruction quality. These results validate that the additional security features do not introduce performance degradation in the primary decoding task.

| Scenes | GS-Hider | | | KeySS (op,ro,sc,xyz) | | |
|---|---|---|---|---|---|---|
| | PSNR↑ | SSIM↑ | LPIPS↓ | PSNR↑ | SSIM↑ | LPIPS↓ |
| Bicycle | 24.377 | 0.735 | 0.254 | 23.333 | 0.596 | 0.424 |
| Bonsai | 31.147 | 0.937 | 0.191 | 31.294 | 0.934 | 0.164 |
| Room | 30.190 | 0.918 | 0.209 | 31.360 | 0.912 | 0.177 |
| Flowers | 20.897 | 0.583 | 0.347 | 19.872 | 0.460 | 0.498 |
| Treehill | 21.952 | 0.625 | 0.361 | 22.510 | 0.580 | 0.449 |
| Garden | 26.954 | 0.855 | 0.118 | 25.683 | 0.757 | 0.284 |
| Stump | 25.565 | 0.731 | 0.259 | 24.548 | 0.656 | 0.409 |
| Counter | 28.053 | 0.899 | 0.202 | 28.433 | 0.887 | 0.189 |
| Kitchen | 29.588 | 0.921 | 0.129 | 30.226 | 0.918 | 0.146 |
| Average | 26.525 | 0.800 | 0.230 | 26.362 | 0.744 | 0.304 |

Table 11: PSNR, SSIM, and LPIPS scores of the pretrained ground-truth 3DGS models used in the proposed method, compared with those used in GS-Hider.

| Cover | $PSNR_c$ ↑ | $SSIM_c$ ↑ | $LPIPS_c$ ↓ | $PSNR_s$ ↑ | $SSIM_s$ ↑ | $LPIPS_s$ ↓ |
|---|---|---|---|---|---|---|
| Bicycle | 23.011 | 0.566 | 0.457 | 29.533 | 0.895 | 0.293 |
| Bonsai | 31.081 | 0.934 | 0.167 | 25.456 | 0.798 | 0.330 |
| Room | 30.785 | 0.906 | 0.192 | 23.877 | 0.657 | 0.389 |
| Flowers | 19.476 | 0.426 | 0.535 | 29.272 | 0.891 | 0.303 |
| Treehill | 22.433 | 0.567 | 0.473 | 22.122 | 0.528 | 0.495 |
| Garden | 25.225 | 0.725 | 0.324 | 28.179 | 0.884 | 0.320 |
| Stump | 23.827 | 0.615 | 0.461 | 29.452 | 0.894 | 0.298 |
| Counter | 28.120 | 0.883 | 0.200 | 20.892 | 0.467 | 0.548 |
| Kitchen | 29.862 | 0.913 | 0.157 | 29.064 | 0.912 | 0.207 |
| Average KeySS | 25.980 | 0.726 | 0.330 | 26.427 | 0.770 | 0.354 |
| Average GS-Hider | 25.817 | 0.782 | 0.246 | 25.178 | 0.780 | 0.306 |

Table 12: All metrics of our method on single secret hiding. Secret scene names are omitted for brevity; see Table 1.

| Scene | KeySS w/o key | | KeySS w/ key | |
|---|---|---|---|---|
| | $PSNR_{cover}$ ↑ | $PSNR_{secret}$ ↑ | $PSNR_{cover}$ ↑ | $PSNR_{secret}$ ↑ |
| Bicycle | 22.880 | 29.669 | 23.011 | 29.533 |
| Bonsai | 31.117 | 24.554 | 31.081 | 25.456 |
| Room | 30.813 | 23.494 | 30.785 | 23.877 |
| Flowers | 19.483 | 29.518 | 19.476 | 29.272 |
| Treehill | 22.407 | 22.477 | 22.433 | 22.121 |
| Garden | 25.254 | 28.582 | 25.225 | 28.179 |
| Stump | 23.908 | 29.106 | 23.827 | 29.452 |
| Counter | 28.189 | 20.811 | 28.120 | 20.891 |
| Kitchen | 29.777 | 29.065 | 29.862 | 29.064 |
| Average | 25.981 | 26.364 | 25.980 | 26.427 |

Table 13: PSNR scores for comparisons between the proposed method with and without user-specific keys. Secret scene names omitted; see Table 1.

| Ratio | Sequentially | | Randomly | |
|---|---|---|---|---|
| | $PSNR_{cover}$ ↑ | $PSNR_{secret}$ ↑ | $PSNR_{cover}$ ↑ | $PSNR_{secret}$ ↑ |
| 5% | 25.980 | 26.427 | 25.588 | 26.102 |
| 10% | 25.980 | 26.427 | 25.163 | 25.750 |
| 15% | 25.980 | 26.427 | 24.714 | 25.341 |
| 25% | 25.980 | 26.418 | 23.739 | 24.408 |

Table 14: Robustness analysis under different pruning methods.

## J ROBUSTNESS OF THE PRUNING

We explore the method's behavior under different Gaussian pruning scenarios, including sequential pruning based on opacity values and random pruning. Sequential pruning removes Gaussians in ascending opacity order, targeting lower-opacity elements first, while random pruning stochastically removes a proportion of Gaussians regardless of opacity. Results in Table 14 suggest reasonable resilience under these modifications, with both pruning strategies showing generally acceptable performance. The steganographic information appears to maintain certain levels of stability even with partial Gaussian removal, indicating potential robustness against structural changes.

## K MULTI-SECRET HIDING SCALABILITY

Our experiments on multi-secret hiding were limited by hardware constraints (one 24-GB RTX-6000 GPU), so we report results for two secrets under one cover. This extension was achieved without any architectural changes, demonstrating the flexibility of our framework. To explore scalability, we trained a 3-secret hiding configuration using the same settings as the 2-secret case. As shown in Table 15 and Figure 10, increasing the number of secrets does not introduce key interference or affect training stability. However, we observed a degradation in reconstruction fidelity due to the shared capacity of the cover model. Future work will address this by exploring deeper decoders, longer or curriculum-based training schedules, increased model capacity, and regularization strategies to improve scalability.

## L SEMANTICAL SIMILAR CLIP PROMPTS

We performed experiments using synonyms and minor word-order variations, with visualizations in Figure. 9. The findings show that semantically similar keys substantially degrade the reconstruction of the cover scene without causing any secret leakage, thereby maintaining robustness against unauthorized access. This effect is due to the sensitivity of CLIP embeddings: even small semantic changes result in distinct embeddings, which our key-controlled decoder correctly interprets as in-

Table 15: Comparison of the average rendering speed and the adaptability of SIBR viewer (Kerbl et al., 2023).

| Type | Scenes | PSNR (Hiding 2 secrects in 1 cover) | PSNR (Hiding 3 secrects in 1 cover) |
|---|---|---|---|
| cover | bicycle | 22.478 | 21.8 |
| secret1 | bonsai | 29.031 | 22.572 |
| secret2 | room | 26.544 | 20.508 |
| secret3 | counter | - | 20.415 |

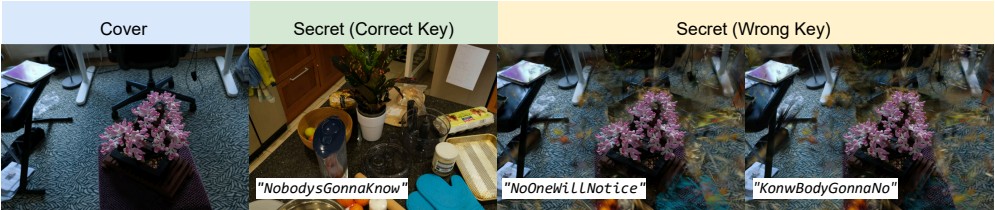

Figure 9: Visualizations of rendering secrets using the correct key and semantically similar incorrect keys (including synonyms and word-order variations)

valid keys. Moreover, we employ 16-character English codes (case-insensitive) as keys, making the likelihood of guessing a semantically similar key extremely low.

## M    MORE VISUALIZATION RESULTS

Comprehensive visualization results are provided in Figure 11 and Figure 12, offering deeper qualitative insights into the method's performance across diverse scene types. Both single and multiple secret hiding scenarios demonstrate exceptional visual fidelity, with the correctly-keyed reconstructions preserving fine geometric details, texture consistency, and color accuracy. When accessed with incorrect keys, the framework demonstrates robust security properties: unauthorized users receive only the cover-scene-like visualization with no discernible traces of the embedded secrets, as evidenced by the absence of visual artifacts or structural inconsistencies that might otherwise suggest hidden content. These visualizations complement the quantitative metrics by confirming the method's practical effectiveness in maintaining the visual-perceptual balance between high-quality secret reconstruction and security.

## N    LLM USAGE STATEMENT

We used large language models for text polishing and grammar correction during manuscript preparation. No LLMs were involved in the design of the method, experiments, or analysis. All content has been carefully verified and validated by the authors.

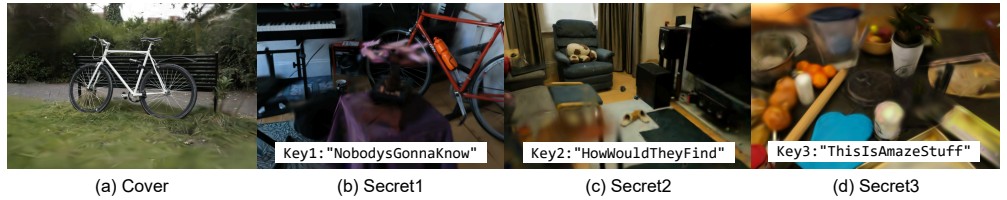

(a) Cover      (b) Secret1      (c) Secret2      (d) Secret3

Figure 10: Visualizations of the results on 3 secret hiding

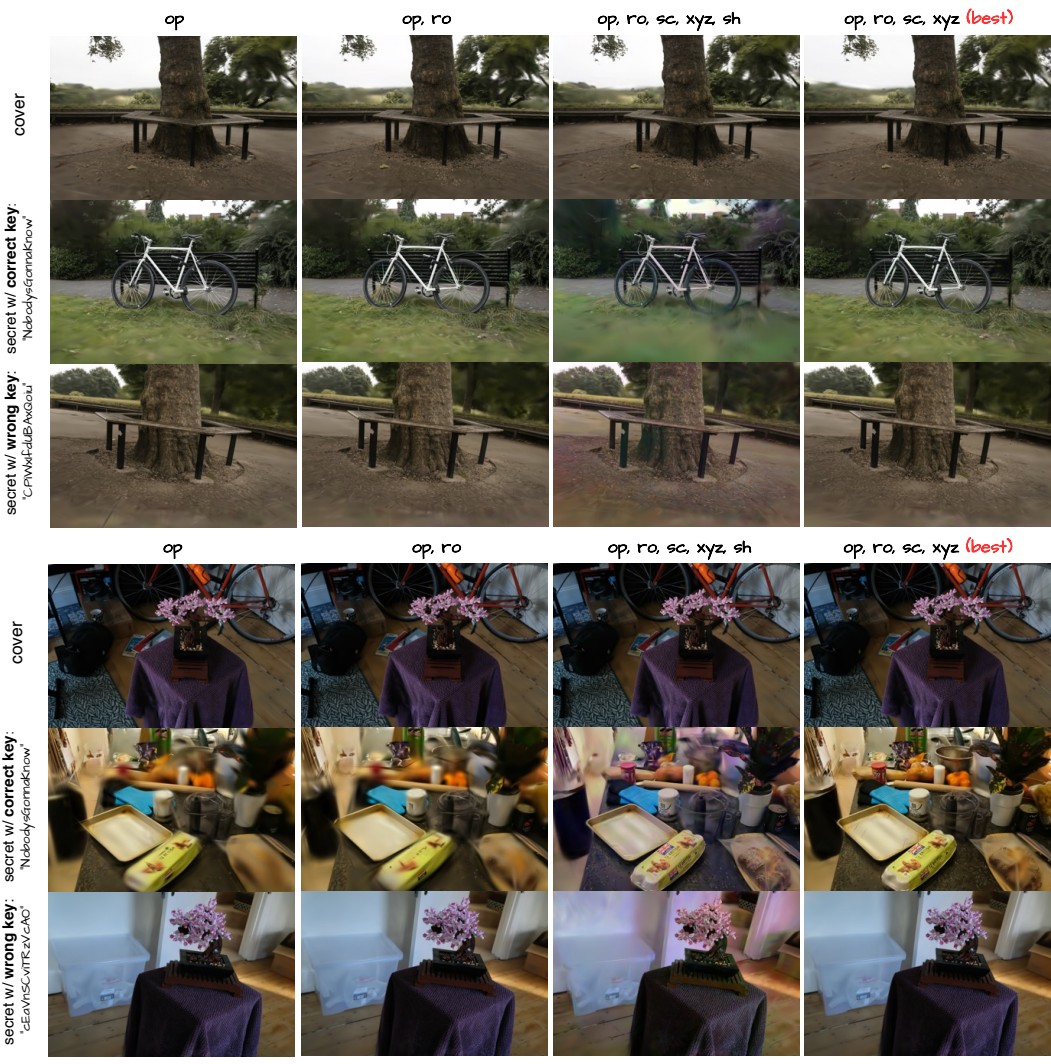

Figure 11: More visualizations of the proposed method on hiding single secert across different attribute update combinations using correct and incorrect keys, which show cover recovery, secret recovery (correct key) and security preservation (incorrect key). Notation follows Figure 4.

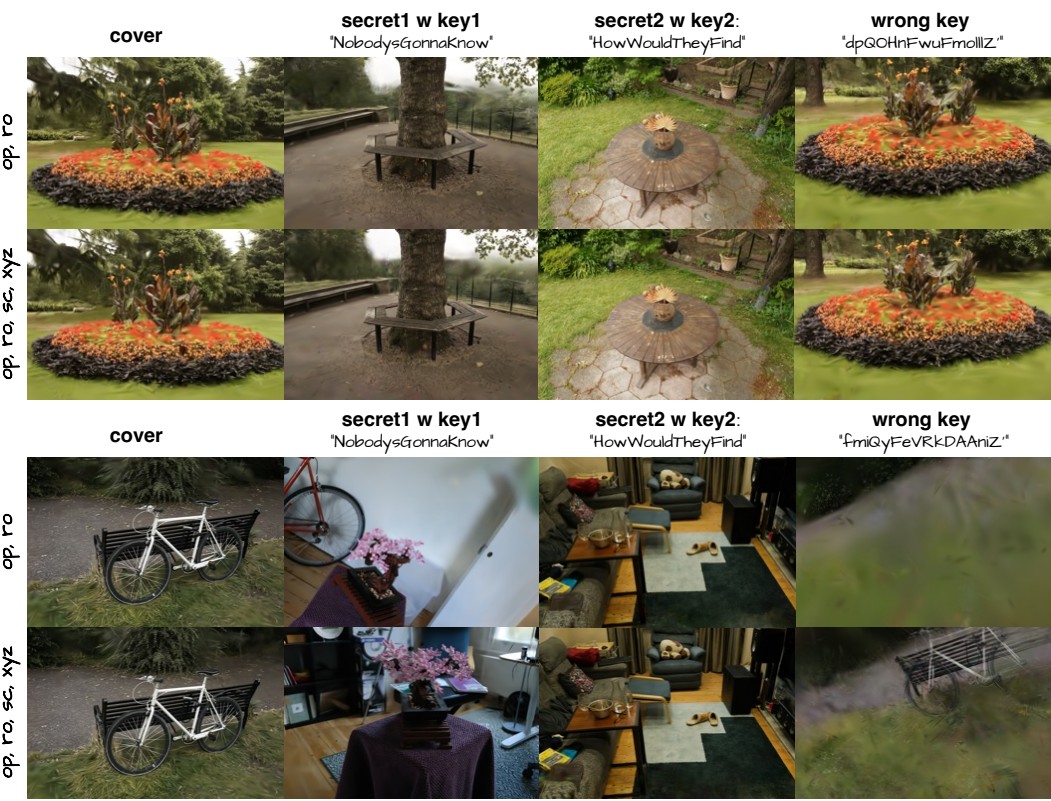

Figure 12: More visualizations of the proposed method on hiding multiple secrets across different attribute update combinations using correct and incorrect keys, which show cover recovery, secret recovery (correct key) and security preservation (incorrect key). Notation follows Figure 4.

