# OpenReview forum: "All That Glitters Is Not Gold: Key-Secured 3D Secrets within 3D Gaussian Splatting"
_ICLR.cc/2026/Conference — ICLR 2026 Poster_

### Official Review · Reviewer_djaB · 2025-10-27

**Soundness:** 3
**Presentation:** 2
**Contribution:** 3
**Rating:** 4
**Confidence:** 4

**Summary:**

This paper proposes KeySS, an end-to-end 3D steganography framework that hides multiple secret 3D scenes inside a cover scene using 3D Gaussian Splatting, while maintaining compatibility with standard rendering pipelines. It introduces a key-controlled decoder to ensure that only the correct key can reveal the hidden secrets, and develops a 3D-Sinkhorn metric to evaluate imperceptibility in the 3D Gaussian space. Experiments show that KeySS achieves high reconstruction fidelity, strong security against unauthorized access, and flexibility for multi-secret hiding.

**Strengths:**

The paper proposed a novel end-to-end framework for 3D steganography, especially for 3D Gaussian Splatting.
- Key-secured decoding architecture is utilized to transfer the original Gaussians to a set of secret Gaussians, which also supports multiple-secret hiding.
- A new metric, the 3D Sinkhorn distance, to measure the distributional disparities between the original and steganographic Gaussians.
- Extensive Experiments on Mip-NeRF360 and Deep Blending, and ablation studies have shown the effectiveness of the proposed method
- The paper is well written, and the method clarification is clear.

**Weaknesses:**

1. **Concept confusion**: The 'features', utilized in Line 015, Line 100, Line 107, etc, should be the 'attributes' of Gaussians. Using the term "feature" may cause confusion with the "feature" is used in GS-Hider.

2. **Motivation should be further clarified**:
- Mentioned in Line 48, the rendering strategy in GS-Hider introduces deviations from the standard GS pipeline. It’s unclear what these deviations refer to, and whether any experiments or visualizations support them. The Keyss method also adds extra MLPs; does this not also deviate from the standard GS pipeline?
- The main motivation of the proposed method is that "Gaussian Attributes
contribute unequally to steganographic effectiveness", while all attributes are equally learned, a transformation from the original Gaussian via a set of MLPs. However, all attributes are transformed equally via a shared set of MLPs. Moreover, Tab. 4 shows that all attributes except 'SH' are used, which seems inconsistent with the stated motivation. Since the method essentially learns a transformation of the secret scene, it is natural that all attributes are required for rendering in a new scene. Therefore, further clarification is needed on how the method design aligns with the proposed motivation.


3. **Clarification of the used metrics**:
- The proposed 3D Sinkhorn distance is less motivated: The Sinkhorn distance differs from the Wasserstein distance (commonly used Optimal Transport) mainly by the addition of an entropy regularization term. A theoretical justification for choosing Sinkhorn over standard OT is needed, as the motivation behind this choice is not clearly explained.
- Main results in Tab. 1 show the effectiveness of the proposed method only by PSNR, while the component selection for the proposed are mainly depends on the proposed Score, shown in Tab. 4. This inconsistent use of the two metrics is confusing. Moreover, the ranking of different settings by Score in Table 4 does not align with their PSNR rankings, which calls for more visual evidence to justify that Score is a more effective metric than PSNR.


4. **Experimental Details**:
- The comparison with related work in the experiments is insufficient (only compared with GS-Hider), though clarified in Line 322 WaterGS and SecureGS are both not open-sourced.
- The input key is embedded with a Clip text encoder. When the input wrong key is semantically similar but not identical to the correct key (e.g., synonyms or slight word order changes), can the model still successfully reconstruct the cover scene? Additionally, the ratio of correct and wrong keys used during training should be clearly described.
- The scales of the scenes in Mip-NeRF360 and DeepBlending differ, resulting in different camera intrinsics and extrinsics. How these camera details are handled should be clearly explained.

5. **Visualization should be refined**:
- Fig.1 and Fig.2 are hard to follow due to low-contrast colors and unclear fonts. The color scheme makes it difficult to distinguish key elements, and the font choice affects readability. Improving visual clarity would make the diagram easier to understand.
- The Visualization in Fig.4, Fig.5 are blurred. The specific resolution settings for MIP-NeRF360 and DeepBlender used in the experiments should be explicitly reported for reproducibility and fair comparison.

**Questions:**

Refer to the weakness section.

**Details Of Ethics Concerns:**

No Ethics Concerns.

---

> ### Author Response · Authors · 2025-11-22
> **Response to Reviewer djaB (1/4)**
>
> Dear Reviewer djaB,
>
> Thank you very much for taking the time to review our manuscript and for offering many valuable comments that helped us improve the work.
>
> We sincerely appreciate your recognition of the novelty and clarity of our proposed KeySS framework. Your acknowledgment of the comprehensive experiments and clear presentation reinforces the impact of our work.
>
> We will address your concerns point by point.
>
> ---
>
> **[W1] Concept confusion: The 'features', utilized in Line 015, Line 100, Line 107, etc, should be the 'attributes' of Gaussians. Using the term "feature" may cause confusion with the "feature" is used in GS-Hider.**
>
> Thank you for pointing this out. We agree that using 'features' for Gaussian attributes may cause confusion. In the revised version, we have **replaced 'features' with 'attributes' when referring to Gaussian properties**, and retain 'features' only when discussing learned embeddings of Gaussians. We believe this distinction enhances the precision and readability of our work. We are grateful for your detailed comment, as it directly contributed to this crucial terminological refinement.
>
> ---
>
> **[W2] Motivation should be further clarified:**
>
> ---
>
> - **[W2-1] Mentioned in Line 48, the rendering strategy in GS-Hider introduces deviations from the standard GS pipeline. It’s unclear what these deviations refer to, and whether any experiments or visualizations support them. The Keyss method also adds extra MLPs; does this not also deviate from the standard GS pipeline?**
>
>   Thank you for the insightful question. We address it by (1) outlining the standard 3DGS rendering pipeline, (2) clarifying the specific deviations introduced by GS-Hider, and (3) explaining why the additional MLPs in KeySS do not constitute a deviation from the standard rendering process.
>
>   **(1) Standard 3DGS rendering pipeline**
>   In the canonical Gaussian Splatting pipeline, each Gaussian is represented by a set of attributes: 3D position (xyz), opacity, spherical harmonics (SH) coefficients, rotation, and scale. These attributes are directly rasterized into the 2D image plane using the standard blending logic, and common WebGL implementations can render such models without any auxiliary modules. Crucially, this pipeline relies solely on these predefined attributes; no external decoders or feature transformations intervene in the per-frame rendering.
>
>   **(2) Deviations introduced by GS-Hider**
>   GS-Hider departs from the standard 3DGS workflow in two ways:
>   - It replaces SH coefficients with learned color features, altering the canonical set of Gaussian attributes.
>   - It introduces a decoder-based rendering stage, meaning that the final RGB values are no longer obtained through the standard 3DGS blending process but instead reconstructed through an additional neural module.
>
>   These attribute-level and renderer-level modifications change both the attribute representation and the rendering behavior, making the resulting pipeline incompatible with standard 3DGS renderers.
>
>   **(3) Why KeySS’s extra MLPs do not deviate from standard rendering**
>   In contrast, KeySS preserves the complete original set of Gaussian attributes and retains the unmodified 3DGS rendering pipeline for all cover-scene rendering. The additional MLPs are not part of the standard rasterization path; they are invoked only during the secret-revealing stage and only for authorized users. As a result, normal users viewing the cover model experience:
>   - unchanged Gaussian attributes,
>   - unaltered rendering logic, and
>   - full compatibility with existing 3DGS/WebGL viewers.
>
>   This design ensures imperceptibility and practicality while still supporting real-time performance.
>
>   If you have further suggestions or concerns, we would be glad to continue the discussion. Your feedback is genuinely helpful in strengthening our work.

---

> ### Author Response · Authors · 2025-11-22
> **Response to Reviewer djaB (2/4)**
>
> - **[W2-2] The main motivation of the proposed method is that "Gaussian Attributes contribute unequally to steganographic effectiveness", while all attributes are equally learned, a transformation from the original Gaussian via a set of MLPs. However, all attributes are transformed equally via a shared set of MLPs. Moreover, Tab. 4 shows that all attributes except 'SH' are used, which seems inconsistent with the stated motivation. Since the method essentially learns a transformation of the secret scene, it is natural that all attributes are required for rendering in a new scene. Therefore, further clarification is needed on how the method design aligns with the proposed motivation.**
>
>   Thank you for the helpful comments. We first clarify a potential misunderstanding regarding our architecture, then explain how our design reflects the unequal contribution of Gaussian attributes to steganographic effectiveness, and finally address the interpretation of Table 4.
>
>   **(1) Clarification on shared vs. attribute-specific layers**
>   Our decoder is not limited to a set of fully shared MLP layers. In addition to the shared embedding layers, it includes **attribute-specific MLP branches** that independently predict each Gaussian attribute. These branches do not share parameters, enabling the model to adjust the learning dynamics for each attribute. This structure is detailed in Section 3.2, Equation 8, and Figure 3(b).
>
>   **(2) Evidence that attributes contribute unequally**
>   To test whether attributes are learned or utilized equally, we analyzed the activation strength of the shared layers for each attribute, as shown in the following table.
>
>   | Attribute | Activation Feature Contribution Percentage |
>   | --------- | ------------------------------------------ |
>   | Opacity   | 72.08%                                     |
>   | Rotation  | 15.80%                                     |
>   | Scale     | 9.82%                                      |
>   | XYZ       | 2.30%                                      |
>
>   These values were computed by summing activation magnitudes over a batch, normalizing them, and reporting the relative ratios. The results clearly show that the attributes are **unequally learned**. This supports our core motivation: **Gaussian attributes do not participate equally in effective secret hiding**.
>
>   **(3) How this model design aligns with the motivation**
>   Motivated by the observation in *Figure 2*, we investigated how updating specific attributes affects steganographic fidelity and leakage. For example, as demonstrated in *Figure 7*, opacity offers the highest hiding fidelity but also poses a risk of secret leakage. Thus, our design aims to identify an optimal strategy for updating Gaussian attributes to achieve **a better fidelity–security tradeoff**.
>
>   **(4) Clarifying Table 4**
>   Our method always **retains the full set of Gaussian attributes** for rendering, ensuring compatibility with the standard 3DGS pipeline. However, during *secret hiding*, the decoder **selectively updates subsets of attributes**. *Table 4* shows that “excluding SH while updating others performs best” reflects an attribute-selection strategy for hiding.
>
>   Please feel free to share any additional suggestions or thoughts. We’d be delighted to discuss them with you.

---

> ### Author Response · Authors · 2025-11-22
> **Response to Reviewer djaB (3/4)**
>
> **[W3] Clarification of the used metrics:**
>
> ---
>
> - **[W3-1] The proposed 3D Sinkhorn distance is less motivated: The Sinkhorn distance differs from the Wasserstein distance (commonly used Optimal Transport) mainly by the addition of an entropy regularization term. A theoretical justification for choosing Sinkhorn over standard OT is needed, as the motivation behind this choice is not clearly explained.**
>
>   Thank you for raising this point. We chose Sinkhorn distance over the Wasserstein distance for two main reasons:
>
>   - **Debiased and Regularized**: Sinkhorn acts as an entropy-regularized approximation to Wasserstein-2, which improves stability and prevents degenerate transport plans during optimization [1]. This regularization ensures convexity and smooth gradients, making the metric robust for high-dimensional Gaussian attributes.
>
>   - **Scalability**: Computing unregularized OT for large point sets (as in 3D Gaussians) is computationally expensive. Sinkhorn enables efficient matrix scaling and faster convergence, making it practical for iterative training and evaluation in high-dimensional settings [2].
>
>   This choice is **consistent with prior work in 3D Gaussian-based tasks**, such as WaSt-3D [3], where Sinkhorn is preferred for its tractability and smoothness. The key difference lies in definition and usage: WaSt-3D defines Sinkhorn divergence as a regularized Wasserstein-2 loss minimized during stylization, whereas our 3D Sinkhorn distance is used as a diagnostic metric to measure distributional shifts between cover and stego models for imperceptibility analysis.
>
>   Thanks again for pointing this out. We have clarified this motivation and added references in the revised paper (*Section 3.3*).
>
>   References:
>
>   [1] Hicham Janati, Marco Cuturi, and Alexandre Gramfort. Debiased sinkhorn barycenters. In International Conference on Machine Learning, pp. 4692–4701. PMLR, 2020.
>
>   [2] Promit Ghosal and Marcel Nutz. On the convergence rate of sinkhorn’s algorithm. Mathematics of Operations Research, 2025.
>
>   [3] Dmytro Kotovenko, Olga Grebenkova, Nikolaos Sarafianos, Avinash Paliwal, Pingchuan Ma, Omid Poursaeed, Sreyas Mohan, Yuchen Fan, Yilei Li, Rakesh Ranjan, et al. Wast-3d: Wasserstein-2 distance for scene-to-scene stylization on 3d gaussians. In European Conference on Computer Vision, pp. 298–314. Springer, 2024
>
>   ---
> - **[W3-2] Main results in Tab. 1 show the effectiveness of the proposed method only by PSNR, while the component selection for the proposed are mainly depends on the proposed Score, shown in Tab. 4. This inconsistent use of the two metrics is confusing. Moreover, the ranking of different settings by Score in Table 4 does not align with their PSNR rankings, which calls for more visual evidence to justify that Score is a more effective metric than PSNR.**
>
>   Thank you for the thoughtful question. We clarify the metric usage and the relationship between PSNR and our proposed Score as follows.
>
>   **(1) Why Table 1 reports only PSNR**
>   Our proposed core requires computing a 3D Sinkhorn distance between the **groundtruth cover** and the **stego cover**. This depends on having access to the stego cover model. Baselines do not provide trained 3DGS assets or decoding models, making the proposed score **impossible to compute** for them. Therefore, following the GS-Hider evaluation protocol, Table 1 compares all methods using PSNR only.
>
>   **(2) Why PSNR and Score rank methods differently**
>   PSNR and Score measure **different aspects** and are not directly comparable as a single “effectiveness” metric.
>
>   - **PSNR** evaluates only *reconstruction fidelity* of the revealed secret.
>   - **Score**, by design, jointly accounts for **fidelity and security**.
>     Its formulation **includes PSNR** but also penalizes configurations that increase leakage or reduce imperceptibility.
>
>   As a result, a setting with high PSNR but poor security can receive a **lower Score**, leading to different rankings between the two metrics.

---

> ### Author Response · Authors · 2025-11-22
> **Response to Reviewer djaB (4/4)**
>
> **[W4] Experimental Details:**
>
> ---
>
> - **[W4-1] The comparison with related work in the experiments is insufficient (only compared with GS-Hider), though clarified in Line 322 WaterGS and SecureGS are both not open-sourced.**
>
>   - Thank you for pointing this out. For WaterGS and SecureGS, we provide a framework-level comparison focusing on imperceptibility, flexibility, and practicality (More details can be found in *Section 4.1)*. This is because they neither are open-sourced and does not provide ground-truth 3DGS models for quantitative evaluation, which makes an implementation-level comparison infeasible.
>   - To address this gap and promote reproducibility, **we have open-sourced KeySS**, making it the first publicly available 3D steganography framework. This will allow future work, including WaterGS and SecureGS, to benchmark against our method.
>   - If the reviewer has a better way to perform a fair comparison with WaterGS and SecureGS, please feel free to share. We would be happy to follow.
>
>   ---
> - **[W4-2] The input key is embedded with a Clip text encoder. When the input wrong key is semantically similar but not identical to the correct key (e.g., synonyms or slight word order changes), can the model still successfully reconstruct the cover scene? Additionally, the ratio of correct and wrong keys used during training should be clearly described.**
>
>   Thank you for the question.
>
>   **(1) Effect of semantically similar but incorrect keys**
>   We conducted **experiments with synonyms and slight word-order variations** and visualized the results in *Figure 9* in *Appendix L*. These results show that semantically similar keys degrade the reconstruction of the cover scene but **do not lead to secret leakage**, ensuring robustness against unauthorized access.
>
>   **(2) Ratio of correct vs. wrong keys during training**
>   During training, we enforce robustness by pairing each correct key with a wrong key in every iteration,resulting in a **1:1 ratio**, which we have explicitly highlighted in the revised paper. This design ensures both security and practical usability (*Appendix C*).
>
>   We hope this addresses both concerns.
>
>   ---
> - **[W4-3] The scales of the scenes in Mip-NeRF360 and DeepBlending differ, resulting in different camera intrinsics and extrinsics. How these camera details are handled should be clearly explained.**
>
>   Thank you for this thoughtful question. We **follow GS-Hider’s configuration** as the benchmark for single-secret hiding, which includes both MipNeRF360 and DeepBlending datasets (*Section 4*). Since the camera intrinsics of these scenes differ (*Section 3.4*), we augment the training dataset by combining camera poses from both the cover and secret scenes.
>   - For poses unique to either scene, we synthesize the corresponding ground-truth images using pre-trained models: the cover 3D model (without embedding) for secret-scene poses, and vice versa.
>   - For shared camera poses, we directly use the pre-trained Gaussian Splatting models on MipNeRF360 and DeepBlending to ensure consistency in geometry and accurate reconstruction across datasets.
>
>   This strategy preserves the integrity of the original camera parameters while enabling robust training under heterogeneous scene configurations.
>
>   ---
>
> **[W5] Visualization should be refined:**
>
> ---
>
> - **[W5-1] Fig.1 and Fig.2 are hard to follow due to low-contrast colors and unclear fonts. The color scheme makes it difficult to distinguish key elements, and the font choice affects readability. Improving visual clarity would make the diagram easier to understand.**
>
>   Thank you for the suggestion. In the updated version, we have improved *Figure 1* and *Figure 2* by adjusting the color scheme and font to enhance contrast and readability, making key elements easier to distinguish. If you feel additional adjustments would further enhance clarity, please let us know. We are happy to refine the figures further.
>
>   ---
> - **[W5-2] The Visualization in Fig.4, Fig.5 are blurred. The specific resolution settings for MIP-NeRF360 and DeepBlender used in the experiments should be explicitly reported for reproducibility and fair comparison.**
>
>   - Thank you for the suggestion. In the revised version, we improved the clarity of *Figure 4* and *Figure 5* by increasing their resolution.
>   - For DeepBlending, the original resolution of 1264 × 832 is used.
>   - For Mip-NeRF360, we use the dataset’s native resolution for normal tests (e.g., "image_4" folder) and higher-resolution images for 4K tests (e.g., "image" folder), noting that resolutions vary slightly across categories. We appreciate the comment, as it helped us clarify these details.

---

### Official Review · Reviewer_noTZ · 2025-10-29

**Soundness:** 4
**Presentation:** 2
**Contribution:** 2
**Rating:** 8
**Confidence:** 3

**Summary:**

This paper introduces KeySS (Key-Secured 3D Steganography), a novel end-to-end framework for hiding 3D secret scenes within a 3D Gaussian Splatting (3DGS) cover scene. The core contributions includes (1) a joint optimization framework that learns the 3DGS cover representation and a key-secured decoder simultaneously, (2) a novel security-enforcing loss function ($\mathcal{L}_{incorrect}$) 1 that trains the decoder to reconstruct the original cover scene when an incorrect key is provided, thus preventing unauthorized access, and (3) a new security evaluation metric, 3D-Sinkhorn distance , which measures the statistical imperceptibility of the steganography directly in the 3D Gaussian parameter space, offering a more robust analysis than traditional 2D render-based metrics.

**Strengths:**

- This work represents a major step forward for 3D steganography. It provides a solution that is not only effective but also practical: it is secure (multi-key, wrong-key defense), high-fidelity, and computationally efficient (maintains 130 FPS rendering). The ability to hide entire 3D scenes (not just watermarks) opens new application possibilities. Furthermore, the 3D-Sinkhon metric is a significant contribution to the community and will likely become a standard for evaluating future work in 3D representation steganography.

-  The KeySS framework is well-designed, end-to-end, and thoughtfully engineered (e.g., using $1 \times 1$ convolutions inspired by PointNet in the decoder). The experimental evaluation is comprehensive and convincing.

**Weaknesses:**

The framework is designed to hide secret 3D scenes within a cover 3D scene ($\mathcal{G}_{cover} \rightarrow \mathcal{G}_{secret}^s$). This is a powerful capability. However, traditional steganography often involves hiding other data types, like text documents, bitstreams, or audio. It is unclear how KeySS would be adapted for this. Would the text file first need to be represented as a 3DGS scene? This seems inefficient. A brief discussion on the framework's adaptability to more conventional steganographic payloads would be beneficial.

**Questions:**

Please refer to Weakness part.

---

> ### Author Response · Authors · 2025-11-22
> **Response to Reviewer noTZ**
>
> Dear Reviewer noTZ,
>
> Thank you very much for taking the time to review our manuscript and for offering many valuable comments that helped us improve the work.
>
> We sincerely appreciate your recognition of the significance of our contributions, including the practicality and security of KeySS (multi-key, wrong-key defense), its high-fidelity and real-time rendering performance, and the introduction of the 3D-Sinkhorn metric as a new standard for evaluating steganographic security. Your acknowledgment of the framework’s thoughtful design and comprehensive experimental validation reinforces the impact of our work.
>
> We will address your concerns point by point.
>
> ---
>
> **[W1] The framework is designed to hide secret 3D scenes within a cover 3D scene ($\mathcal{G}\_{cover} \rightarrow \mathcal{G}\_{secret}^s$). This is a powerful capability. However, traditional steganography often involves hiding other data types, like text documents, bitstreams, or audio. It is unclear how KeySS would be adapted for this. Would the text file first need to be represented as a 3DGS scene? This seems inefficient. A brief discussion on the framework's adaptability to more conventional steganographic payloads would be beneficial.**
>
> Thank you for raising this point. While KeySS is primarily designed for 3D scene steganography, its adaptability to other data types is promising. A straightforward approach is to **leverage the multi-view nature of 3DGS**: each rendered view can embed 2D steganographic payloads (e.g., text, bitstreams, or audio), effectively transforming the problem into image steganography while benefiting from the large capacity offered by multiple views. This significantly increases the embedding bandwidth compared to single-image methods. We plan to investigate this in future work.

---

> > ### Comment · Reviewer_noTZ · 2025-11-26
> >
> > Thanks for the authors' response. Currently I have no more questions, and I will keep my score.

---

> ### Author Response · Authors · 2025-11-26
> **Appreciation for the Positive Feedback from Reviewer noTZ**
>
> Thank you very much for keeping your positive rating and for your encouraging feedback. We truly appreciate your support.

---

### Official Review · Reviewer_syDs · 2025-11-01

**Soundness:** 3
**Presentation:** 3
**Contribution:** 3
**Rating:** 6
**Confidence:** 4

**Summary:**

This paper introduces KeySS, a novel key-secured 3D steganography framework built upon 3D Gaussian Splatting (3DGS) for hiding 3D secrets within 3D scenes. Unlike prior 3D steganographic methods that rely on suboptimal Gaussian features and risk detectability, KeySS jointly optimizes a 3DGS model and a key-protected decoder to balance reconstruction fidelity and steganographic security. The framework enables multi-secret hiding and prevents unauthorized access via a controllable key mechanism. It also proposes 3D-Sinkhorn distance analysis, a new metric to quantitatively evaluate imperceptibility by comparing Gaussian distributions between normal and steganographic scenes. Extensive experiments demonstrate that KeySS achieves state-of-the-art reconstruction quality and strong security guarantees, while remaining efficient and scalable for multi-GPU training.

**Strengths:**

1. End-to-End 3D Steganography Framework: The paper proposes KeySS, an end-to-end 3D steganographic learning framework that jointly optimizes 3D Gaussian representations and a key-secured decoder, ensuring both high-fidelity rendering and strong steganographic compatibility with the standard 3DGS pipeline.

2. Key-Secured Decoder for Multi-Secret Recovery: A key-controllable decoding mechanism is introduced to enable secure and accurate reconstruction of multiple hidden 3D secrets, effectively preventing unauthorized access.

3. Fidelity–Security Tradeoff Analysis and New Metric: The authors systematically explore how different 3D Gaussian feature combinations affect fidelity and imperceptibility, and propose 3D-Sinkhorn distance, a novel quantitative metric for evaluating steganographic security beyond traditional 2D measures.

**Weaknesses:**

1. Although KeySS enhances security, its rendering speed is slower than standard 3DGS, which may limit its applicability in real-time scenarios.

2. The authors should provide more illustrative examples and application scenarios to better explain the necessity of introducing key-control, thereby enhancing the motivation, significance, and practical relevance of the paper.

3. The improvement in rendering quality over GS-Hider is relatively limited, while the performance degradation compared to standard 3DGS is significant.

**Questions:**

Please refer to the weakness.

---

> ### Author Response · Authors · 2025-11-22
> **Response to Reviewer syDs**
>
> Dear Reviewer syDs,
>
> Thank you very much for taking the time to review our manuscript and for offering many valuable comments that helped us improve the work.
>
> We sincerely appreciate your recognition of our contributions, including the end-to-end 3D steganography framework, the key-secured multi-secret recovery mechanism, and the fidelity–security tradeoff analysis with the proposed 3D-Sinkhorn metric. Your acknowledgment of these aspects reinforces the significance and practical relevance of our work.
>
> We will address your concerns point by point.
>
> ---
>
> **[W1] Although KeySS enhances security, its rendering speed is slower than standard 3DGS, which may limit its applicability in real-time scenarios.**
>
> We appreciate the reviewer’s comment and the opportunity to clarify this point. As shown in Table 9, KeySS achieves **the same rendering speed as standard 3D Gaussian Splatting**, maintaining real-time performance of approximately **130 FPS**. This is achieved for two main reasons:
> - **Preserved pipeline**: The standard Gaussian Splatting rendering process is fully retained.
> - **Lightweight decoder**: The key-controlled decoder consists of lightweight MLP layers, introducing only negligible overhead.
>
> We thank the reviewer for allowing us to clarify this aspect of our rendering speed.
>
> ---
>
> **[W2] The authors should provide more illustrative examples and application scenarios to better explain the necessity of introducing key-control, thereby enhancing the motivation, significance, and practical relevance of the paper.**
>
> Key-control plays a pivotal role in practical scenarios, particularly in environments requiring multi-user access management. For example, in a gaming context, players can unlock unique secret scenes embedded within the same cover 3D environment by discovering personalized keys during gameplay. This approach not only **enhances immersion** but also **ensures privacy** by restricting access to individualized content. Furthermore, it supports privacy and flexibility, **enabling diverse user experiences without duplicating assets**. Such mechanisms are highly relevant for applications like personalized AR/VR experiences, secure distribution of 3D assets, and controlled access in cloud-based rendering pipelines. In the revised paper, we have elaborated on these application scenarios to highlight their significance and potential impact (*Section 1*).
>
> ---
>
> **[W3] The improvement in rendering quality over GS-Hider is relatively limited, while the performance degradation compared to standard 3DGS is significant.**
>
> While hiding secrets inevitably introduces some performance degradation compared to standard 3DGS, our method is designed to **maintain real-time rendering without exposing hidden content.** Unlike GS-Hider, which requires modifications to the rendering pipeline that can reveal the presence of hidden data, KeySS preserves the original 3DGS rendering process and attribute formats, ensuring both security and practicality. Furthermore, our approach achieves competitive visual quality while supporting multiple secrets in a single cover scene, offering a better balance between fidelity, efficiency, and robustness than GS-Hider.

---

### Official Review · Reviewer_6JSy · 2025-11-01

**Soundness:** 3
**Presentation:** 3
**Contribution:** 2
**Rating:** 4
**Confidence:** 3

**Summary:**

This paper presents KeySS, a 3D scene steganography framework built upon 3D Gaussian Splatting (3DGS). The method embeds complete secret scenes within a cover representation while maintaining high visual fidelity and compatibility with standard 3DGS rendering. A key-conditioned decoder, guided by CLIP-based semantic embeddings, modulates selected Gaussian attributes to securely encode and decode hidden content. To evaluate imperceptibility, the authors introduce a 3D-Sinkhorn distance that quantifies subtle distributional deviations in Gaussian parameter space.The paper’s main strength lies in its innovative integration of key-controlled steganography into 3D Gaussian Splatting, achieving secure multi-scene hiding with high fidelity and full rendering compatibility.

**Strengths:**

The paper’s main strength lies in its innovative integration of key-controlled steganography into 3D Gaussian Splatting, achieving secure multi-scene hiding with high fidelity and full rendering compatibility.

**Weaknesses:**

Please see questions.

**Questions:**

Technically, the paper adapts fundamental ideas from information security and steganography to the 3DGS framework. It incorporates a key-controlled access mechanism into a neural decoder, where CLIP text embeddings act as learnable semantic keys that determine whether the model reconstructs the hidden scene or reproduces the visible cover. The classical concept of secure hiding, which reveals the secret only with the correct key and restores the cover otherwise, is reinterpreted in the 3D Gaussian parameter space by modulating attributes such as opacity, rotation, and position instead of image pixels. In addition, the paper extends the notion of distributional detectability from traditional steganalysis to 3D by introducing the 3D-Sinkhorn distance, a metric designed to measure subtle distributional variations among Gaussian parameters. Together, these components represent a careful transfer of established security mechanisms into modern 3D scene representation learning.

The method jointly optimizes the cover and multiple secret scenes using a shared decoder, which raises potential concerns about overfitting among different supervision signals. It is not explicitly stated whether the reported quantitative results are obtained on training views or unseen novel views. Since the model operates on 3D Gaussian representations, it would be important to clarify this evaluation setting and, if possible, include visualizations of the reconstructed geometry (for example, depth maps or mesh) to verify consistency.

In addition, The paper demonstrates a two-secret configuration under a single cover but does not discuss scalability beyond this case. It would be valuable for the authors to clarify potential challenges in extending the framework to N > 2 secrets, such as interference between keys, degradation of reconstruction fidelity, or training instability.

---

> ### Author Response · Authors · 2025-11-22
> **Response to Reviewer 6JSy (1/2)**
>
> Dear Reviewer 6JSy,
>
> Thank you very much for taking the time to review our manuscript and for offering many valuable comments that helped us improve the work.
>
> We sincerely appreciate your recognition that our inovative intergration of key-controlled steganography into 3D GS and our good performance on multi-scene hiding.
>
> We will address your concerns point by point.
>
> ---
>
> **[Q1] The method jointly optimizes the cover and multiple secret scenes using a shared decoder, which raises potential concerns about overfitting among different supervision signals.**
>
> Thank you for raising this concern. We interpret your question as asking whether, when multiple secrets (e.g., A and B) are embedded into a single cover, the **shared decoder might overfit to one secret**, causing it to reconstruct secret A correctly but fail to reconstruct secret B (or vice versa).
>
> - Our experiments suggest that this issue does not arise. As shown in *Table 2* and *Figure 5*, the shared decoder reconstructs **all secrets with consistently high PSNR**, indicating that it does not collapse or overfit to any single supervision signal.
>
> - This stability mainly comes from our **key-conditioned design**. The key provides a strong, discriminative conditioning signal that directs the decoder to the correct secret. This setting is analogous to **class- or text-conditioned generative models**, where a single shared generator is trained on many class/text–image pairs. Despite sharing parameters, such models reliably produce different outputs for different conditioning inputs (e.g., “cat” vs. “dog”), because the conditioning signal (class, text, or **in our case, the key**) clearly separates tasks and prevents overfitting to any single class.
>
> - Similarly, in our framework, the key embedding steers the decoder toward the correct secret and prevents the features from collapsing onto a particular scene. This conditioning mechanism effectively mitigates the overfitting concern.
>
> We have incorporated these discussions into Appendix D to facilitate the reader’s understanding. We hope this explanation clarifies why sharing the decoder across multiple secrets does not introduce overfitting issues.
>
> ---
>
> **[Q2] It is not explicitly stated whether the reported quantitative results are obtained on training views or unseen novel views. Since the model operates on 3D Gaussian representations, it would be important to clarify this evaluation setting and, if possible, include visualizations of the reconstructed geometry (for example, depth maps or mesh) to verify consistency.**
>
> Thank you for this question.
> - Our reported results are on **unseen novel views**, following the official Mip-NeRF360 test split and GS-Hider’s evaluation protocol. Specifically, every 8th frame along the 360° trajectory is reserved for testing, while the remaining frames are used for training. A more detailed breakdown of training and testing view **counts** is provided in the table below. (**details added** in *Section 4* and summarized in *Table 8* in *Appendix C*).
>
>     |Rendered_Cover_Scene Rendered_Secret_Scene|Bicycle Playroom|Bonsai Counter|Room Garden|Flowers Playroom|Treehill Bicycle|Garden Playroom|Stump Playroom|Counter Bicycle|Kitchen Bonsai|Average|
>     |-|-|-|-|-|-|-|-|-|-|-|
>     |**Training_Views_Cover**|255|255|272|151|123|161|109|210|169|**189.4**|
>     |**Training_Views_Secret**|196|210|161|196|169|196|196|169|255|**194.2**|
>     |**Novel_Testing_Views_Cover**|25|37|39|22|18|24|16|30|25|**26.2**|
>     |**Novel_Testing_Views_Secret**|29|30|24|29|25|29|29|25|37|**28.6**|
>
> - To address geometric consistency, we provide **depth map visualizations** of our reconstructions in *Figure 8* in *Appendix E*. Additionally, we will release all trained checkpoints on GitHub to ensure reproducibility and allow independent verification.
>
> Thank you again for your helpful feedback, which has allowed us to further improve the completeness and clarity of our work. Please feel free to let us know if you have further suggestions or thoughts.

---

> ### Author Response · Authors · 2025-11-22
> **Response to Reviewer 6JSy (2/2)**
>
> **[Q3] In addition, The paper demonstrates a two-secret configuration under a single cover but does not discuss scalability beyond this case. It would be valuable for the authors to clarify potential challenges in extending the framework to N > 2 secrets, such as interference between keys, degradation of reconstruction fidelity, or training instability.**
>
> Thank you for raising this valuable point. Following your suggestion, we trained a **3-secret** configuration using the same settings as the 2-secret experiment. As shown in *Table 15* and *Figure 10* (Appendix K), increasing the number of secrets leads to **degradation in reconstruction fidelity**, likely due to the small parameter budget of our MLPs and the shared capacity of the cover model.
>
> Importantly, we observed **no key interference** (e.g., key A never reconstructs secret B or C, as shown in Figure 10 in the updated version) and **no training instability** (e.g., losses remained stable and convergent).
>
> Our future work will focus on addressing this by exploring deeper decoders, longer or curriculum-based training schedules, increased model capacity, and regularization strategies to improve scalability.
>
> We appreciate this thoughtful suggestion. It guides us to clarify scalability considerations and highlight future research directions. If you have further feedback, we would be glad to discuss it.

---

### Author Response · Authors · 2025-12-02
**Summary of Our Rebuttal**

Dear Reviewers, ACs, SACs, and PCs

Thank you for your time and dedication. We recognize that the workload is especially heavy this year. To help streamline the discussion, we provide the following summary, which we hope will help the ACs quickly navigate the key issues addressed in our rebuttal and facilitate a smoother evaluation process.

**Strength:** As the reviewer noted, our key-controlled end-to-end 3D steganography framework supports multi-secret hiding with full 3DGS rendering compatibility **(6JSy, syDs)**, high fidelity **(6JSy,noTZ)**, and strong security **(6JSy, syDs, noTZ)**. We introduce a 3D Sinkhorn distance to systematically examine how Gaussian attribute combinations affect fidelity and imperceptibility **(syDs, noTZ, djaB)**. The experimental evaluation is comprehensive and convincing **(noTZ, djaB)**.

---
Please find below a summary of the points raised by the reviewers and how we responded to them **(Revisions in the paper have been highlighted in blue)**.

**1. Response to reviewer 6JSy**

- [Q1] Decoder stability: Key-conditioned decoding prevents overfitting across multiple secrets.

- [Q2] Evaluation protocol transparency: All results use unseen novel views per Mip-NeRF360/GS-Hider protocol. Depth-map visualizations confirm geometric consistency.

- [Q3] Scalability: New 3-secret experiments show no key interference or training instability.

---
**2. Response to reviewer syDs**

- [W1] Rendering speed: KeySS preserves the standard 3DGS pipeline at ~130 FPS.

- [W2] Key-control applications: Highlighted practical applications demonstrating real-world relevance.

- [W3] Baseline comparison: KeySS achieves a stronger balance of fidelity, security, and practicality than GS-Hider by keeping the rendering pipeline unmodified and protecting hidden content.

---
**3. Response to reviewer noTZ**

- [W1] Adaptability to non-3D payloads: We clarified that KeySS can naturally support conventional payloads (text, bitstreams, audio) by benefiting from the large capacity offered by multiple views.

**As a result, reviewer 6JSy has confirmed an overall score of 8.**

---
**4. Response to reviewer djaB**

- [W1] Terminology: Replaced ‘features’ with ‘attributes’ for Gaussian properties.

- [W2] Motivation: Clarified differences in GS-Hider (attribute- and renderer-level changes) versus KeySS. Explained the decoder’s use of both attribute-specific and shared MLPs.

- [W3] Metrics: Justified 3D Sinkhorn distance and demonstrated that our Score jointly measures fidelity and security.

- [W4] Experiments: Open-sourced KeySS, tested robustness against semantically similar keys, and clarified adherence to benchmark protocols.

- [W5] Visualization: Enhanced figure clarity and contrast.

---
Thank you very much!

The Authors

---

### Meta-Review · Area_Chair_pL1U · 2026-01-05

**Summary:**

The paper received mixed reviews, with two positive and two negative assessments. Reviewers 6JSy and djaB raised a range of concerns, many of which were addressed during the rebuttal. However, the issue of potential secret leakage does not appear to be fully resolved and may represent a critical limitation for practical deployment. As discussed in prior work, such as SecureGS, even simple point cloud visualizations can expose sensitive information. Furthermore, the proposed approach appears incompatible with Gaussian pruning, a widely used and essential technique for significantly reducing representation size.

However, the authors have addressed the majority of the raised concerns, and I believe the current contribution is meaningful to the research community. Considering the improvements made during the rebuttal and the potential value of the work, I am inclined to recommend acceptance and I recommend that the authors include a clearer discussion of the above limitations in the final version.

**Reviewer Concerns:**

6JSy: Raised concerns regarding the use of a shared decoder, the experimental setup (training vs. testing views), geometry visualization (depth vs. mesh), and scalability to more than two secrets. I think these issues were addressed during the rebuttal.

syDs: Questioned the claim of improved rendering speed, the motivation for key-controlled secrets, and the marginal performance gains. While the latter two points were partially addressed in the rebuttal, they remain somewhat debatable. Overall, this reviewer maintained a positive assessment of the paper.

noTZ: Asked about extending the approach to other data modalities. This concern appears largely out of scope for the current work, and the reviewer remained supportive of acceptance.

djaB:
a. Terminology: Issues related to confusing terminology were addressed in the rebuttal.

b. Motivation / Security: The authors partially addressed the motivation; however, the risk of secret leakage remains a significant concern. As demonstrated in prior work such as SecureGS, simple point cloud visualizations can already reveal sensitive information. This appears to be an inherent limitation of approaches that rely on standard 3DGS representations and rendering pipelines.

c. 3D Sinkhorn Distance: The response did not fully resolve this concern, though it may not be a critical issue.

d. Missing Comparisons: The absence of comparisons to WaterGS and SecureGS is acceptable given that these methods are not publicly available.

e. Camera Pose Differences Across Scenes: Adequately addressed.

f. Similar-Key Robustness: Concerns regarding performance degradation under similar but incorrect keys remain unresolved. As shown in Figure 9, using a wrong but similar key leads to significant degradation of the cover scene.

g. Visualization and Presentation: Adequately addressed.

**Reviewer Scores:**

See the above sections.

---

### Decision · Program_Chairs · 2026-01-26

Accept (Poster)